# Simulation of extreme rainfall and streamflow events in small Mediterranean watersheds with a one-way coupled atmospheric-hydrologic modelling system

Corrado Camera[1], Adriana Bruggeman[2], George Zittis[3], Ioannis Sofokleous[2], and Joël Arnault[4]

[1] Dipartimento di Scienze della Terra 'A. Desio', Università degli Studi di Milano, Milano, 20133, Italy
[2] Energy Environment and Water Research Center, The Cyprus Institute, Nicosia, 2121, Cyprus
[3] Climate and Atmosphere Research Center, The Cyprus Institute, Nicosia, 2121, Cyprus
[4] Institute of Meteorology and Climate Research, Karlsruhe Institute of Technology, Garmisch-Partenkirchen, 82467, Germany

*Correspondence to*: Corrado Camera (corrado.camera@unimi.it)

**Abstract.**

Coupled atmospheric-hydrologic systems are increasingly used as instruments for flood forecasting and water management purposes, making the performance of the hydrologic routines a key indicator of the model functionality. This study's objectives were: (i) to calibrate the one-way coupled WRF-hydro model for simulating extreme events in Cyprus with observed precipitation; and (ii) to evaluate the model performance when forced with WRF-downscaled ($1 \times 1$ km$^2$) re-analysis precipitation data (ERA-Interim). This set up resembles a realistic modelling chain for forecasting applications and climate projections. Streamflow was modelled during extreme rainfall events that occurred in January 1989 (calibration) and November 1994 (validation) over 22 mountain watersheds. In six watersheds, Nash-Sutcliffe Efficiencies (NSE) larger than 0.5 were obtained for both events. The WRF-modelled rainfall showed an average NSE of 0.83 for January 1989 and 0.49 for November 1994. Nevertheless, hydrologic simulations of the two events with the WRF-modelled rainfall and the calibrated WRF-Hydro returned negative streamflow NSE for 13 watersheds in January 1989 and for 18 watersheds in November 1994. These results indicate that small differences in amounts or shifts in time or space of modelled rainfall, in comparison with observed precipitation, can strongly modify the hydrologic response of small watersheds to extreme events. Thus, the calibration of WRF-Hydro for small watersheds depends on the availability of observed rainfall with high temporal and spatial resolution. However, the use of modelled precipitation input data will remain important for studying the effect of future extremes on flooding and water resources.

## 1 Introduction

Atmospheric and hydrologic processes are strictly related, since they share the land surface as a common interface for moisture and heat fluxes. Precipitation is the primary cause of all surface hydrologic processes, such as overland, subsurface and river flow. Conversely, soil moisture and surface water distributions affect near surface atmospheric conditions and processes, such as the temperature distribution, the structure of the atmospheric boundary layer, the formation of shallow clouds and precipitation amounts (Lin and Cheng, 2016; Zittis et al., 2014 and references therein). In recent years, the scientific community has made ever-increasing efforts to improve the simulation skills of both atmospheric and hydrologic models, leading also to the development of coupled modelling systems. Since the beginning of the 21$^{st}$ century, the main research interest in developing such models has been the evaluation of the feedbacks between the hydrologic cycle and the atmospheric processes, to get a deeper understanding of regional climate change and its impacts (Ning et al., 2019). However, recently authors have started to see these systems as instruments for flood forecasting, making the performance of the hydrologic routines a key indicator of the model quality (Givati et al., 2016; Maidment, 2017).

The Weather Research and Forecasting hydrologic modeling system WRF-Hydro (Gochis et al., 2015) is an example of such a modelling system. It consists of a set of routines extending the hydrologic physics options in the Noah Land Surface Model (Noah LSM, Ek et al., 2003) and Noah with Multi-Parameterization Land Surface Model (Noah-MP LSM, Niu et al., 2011),

which are the most commonly used land surface schemes of WRF (Constantinidou et al., 2019; Skamarock and Klemp, 2008). In relation to WRF, WRF-Hydro can be run in an uncoupled (one-way coupled) mode or in a fully-coupled (two-way coupled) mode. In the first case, WRF-Hydro is run with user's specified atmospheric forcing, which can be observations, reanalyses,

previously calculated model outputs or a mixture of the three (e.g., observed precipitation and WRF-derived temperature, wind speed, humidity, radiation etc). As a result, hydrologic outputs are influenced by the atmospheric variables but not vice versa. In the second case, WRF-Hydro enhanced hydrologic routines update the land surface states and fluxes in the LSM grid, which are then used by the atmospheric component of the model.

As summarized by Rummler et al. (2019), WRF-Hydro is mainly used in its uncoupled mode for model calibration and flood

forecasting (e.g., Lahmers et al., 2019; Maidment, 2017; Silver et al., 2017; Verri et al., 2017; Givati et al., 2016; Yucel et al., 2015). Conversely, the fully-coupled mode is usually adopted to investigate land-atmosphere feedbacks (Arnault et al., 2016, 2019; Rummler et al., 2019; Senatore et al., 2015; Wehbe et al., 2019; Zhang et al., 2019).

Focusing on the use of the model for the simulation of flood events, Yucel et al. (2015) calibrated WRF-Hydro over one watershed and two heavy rainfall events in northern Turkey, using 4-km WRF rainfall as input. The calibrated model

parameters were then applied to three other watersheds and 10 heavy rainfall events. Their main aim was to quantify the performance improvement of the calibrated WRF-Hydro model against its use with default parameterization and test parameter transferability. In addition, they tested the model with WRF, WRF with data assimilation, and EUMETSAT precipitation derived input. They obtained the best results with the calibrated model, forced by WRF with data assimilation precipitation. They suggest that this model configuration allows parameter transferability to ungauged catchments.

Givati et al. (2016) calibrated uncoupled WRF-Hydro based on gridded observations of two high intensity rainfall events that occurred in 2013 over the Ayalon basin in Israel. The calibrated model was subsequently run with WRF-derived precipitation resulting from both uncoupled and fully-coupled simulations. The study demonstrated that both precipitation and streamflow as derived from the fully-coupled model were superior to one-way coupled results, suggesting a possible application of fully coupled systems for early flood warning applications. Still, the authors suggested further research with a similar study set-up

but over areas characterized by different precipitation and hydrologic regimes.

Silver et al. (2017) focused on five extreme events occurring over seven watersheds located in Israel and Jordan. They proposed a procedure for parameterizing the model scaling coefficients related to infiltration partitioning and soil hydraulic conductivity, as well as for defining topographic categories. The procedure was based on soil physical properties and terrain characteristics only. They demonstrated that their method leads to better streamflow predictions than trial and error calibration and is as good

as expert knowledge parameterization.

Verri et al. (2017) calibrated an uncoupled WRF/WRF-Hydro modelling system over the Ofanto river basin, in southern Italy. Focus was on two three-month periods, each characterized by a heavy rainfall event and covering different seasons. WRF was run with 16-km horizontal resolution and 6-h fields forced by ECMWF-IFS (European Centre for Medium-Range Weather Forecasts – Integrated Forecasting System) as initial and boundary conditions. In addition, they presented a WRF rainfall

correction approach based on rainfall observations, an objective analysis and a least square melding scheme and demonstrated that it improved river discharge simulation. The study also showed that optimal, calibrated values of infiltration partitioning and baseflow coefficients differ in the two events, suggesting a seasonal dependence.

Nowadays, uncoupled WRF-Hydro is the core of the National Water Model (NWM, https://ral.ucar.edu/projects/supporting-the-noaa-national-water-model), running over the Conterminous United States and furnishing streamflow forecasts for 2.7

million river reaches. The NWM flood forecasting skills has been strengthened within the framework of the National Flood Interoperability Experiment (Maidment, 2017). The NWM and WRF-Hydro remain under constant development. An example is the study of Lahmers et al. (2019), who added channel infiltration processes to the modelling system to improve streamflow simulations in the arid southwestern United States.

From this review, it appears that few studies focus on the evaluation of the hydrologic output of WRF-Hydro when forced with observed rainfall and just a few more when forced with modelled rainfall. Model performance loss due to differences between observed and modelled rainfall is rarely discussed. Also, little attention has been given to small watersheds (area below 100 km$^2$), which are often ungauged and prone to flash floods. This study aims to address this gap. The focus is on two extreme events that occurred over 22 small watersheds, located in the Troodos Mountains of Cyprus, between 8-10 January 1989 and 20-22 November 1994. The main objectives are: (i) to calibrate the uncoupled WRF-Hydro model for simulating extreme events in Cyprus with observed precipitation; and (ii) to evaluate the model performance when forced with WRF-downscaled ($1 \times 1$ km$^2$) re-analysis precipitation data (ERA-Interim). The model runs covered two 15-day periods (1-16 January and 11-26 November) to include a short spin-up of the WRF-Hydro routines and the simulation and evaluation of the receding limb of the hydrograph.

## 2 Study area

This study focuses on 22 watersheds located on the northern slope of the Troodos Mountains, Cyprus (Figure 1). The bedrock geology of the region is characterized by an ophiolitic complex. The highest peak of Troodos is Mt. Olympus (1952 m a.sl.). At high elevations (above 1400 m a.s.l.), ultramafic rocks are the dominant lithology (harzburgite, serpentinite, pyroxenite, wehrlite and dunite). Moving downhill, dominant rock types show a transition from gabbro to diabase, pillow lavas and sedimentary formations, therefore stratigraphically from the lower to the higher lithotype. Between gabbro and pillow lavas, diabase is present in the form of sheeted dykes and it constitutes the largest area of Troodos outcrop. Often, pillow lavas and sheeted dykes do not present a net geological limit, but the oldest lavas host the youngest dykes (Cleintaur et al., 1977). This transitional zone between pillow lavas and dykes takes the name of basal group. Throughout the ophiolitic complex, bedrock is usually found at shallow depths. According to the digital soil map of Cyprus (Camera et al. 2017), most of the soils over Troodos are Lithic Leptosols with a stony gravelly texture and a predominant very shallow depth (0-10 cm), which can sometimes reach up to 100 cm. These characteristics highlight why rock fractures can be considered the main controlling factor for the region's subsurface hydrology.

Due to its characteristic Mediterranean climate, more than 90% of a hydrologic year's (October-September) runoff from Troodos is produced between December and April. During the summer months, most rivers are completely dry (Le Coz et al., 2016). Due to their small areas and steep slopes, all watersheds have quite short times of concentration. Therefore, intense rainfall events lasting few hours can easily cause floods in the downstream plains.

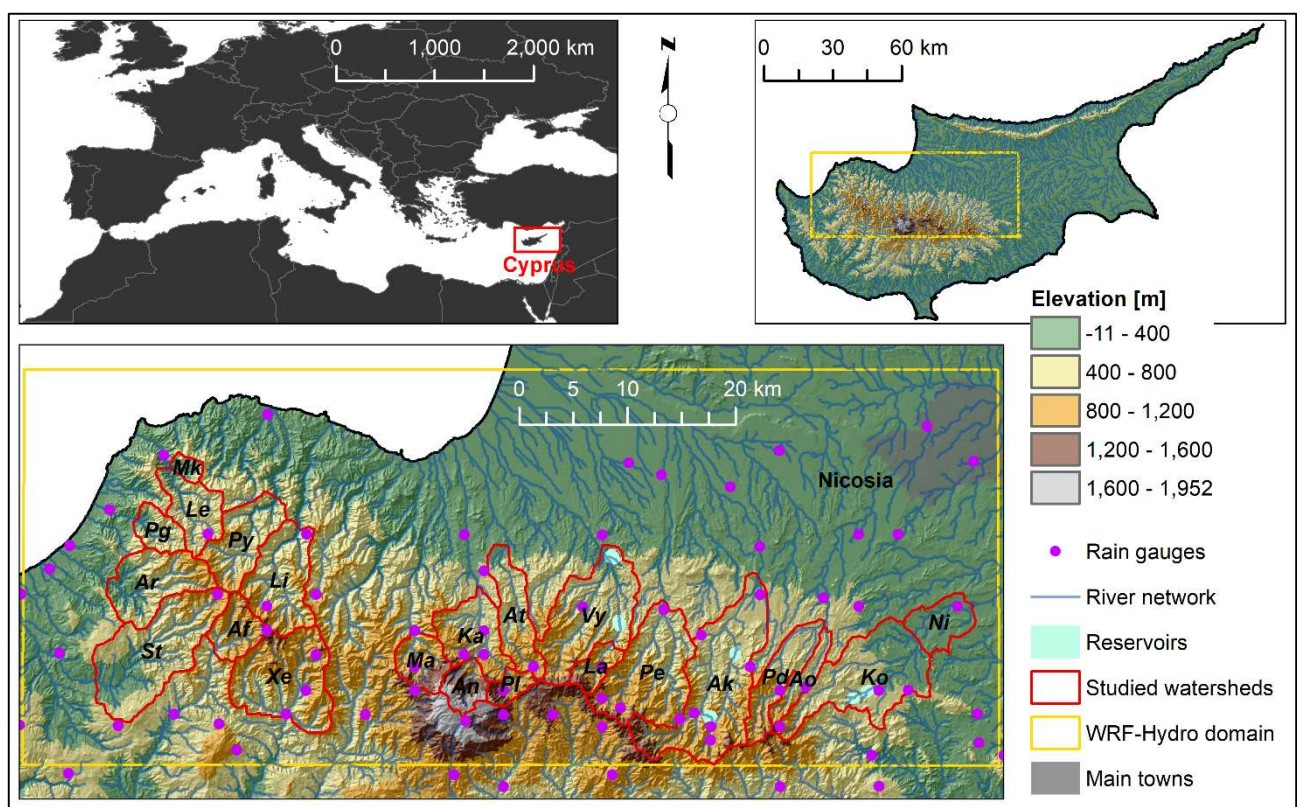

**Figure 1. Geographical setting of the island of Cyprus and WRF-Hydro study area with the 22 target watersheds. For watershed short names refer to Table 1.**

Table 1 lists the 22 watersheds, their area and the total modeled stream length, and summarizes their geology, as obtained from the geological map of Cyprus (Cyprus Geological Survey Department, 1995). Agios Nikolaos and Platania are sub-watersheds of Kargiotis; Lagoudera is a sub-watershed of Vyzakia; Kotsiati is a sub-watershed of Nisou.

**Table 1. Morphological and geological characteristics of the studied watersheds.**

| Watershed | Watershed short name | Area [km²] | Channel length [km] | Ultramafic complex [%] | Gabbro [%] | Sheeted Dikes [%] | Basal group [%] | Pillow Lavas [%] | Sedimentary formations [%] |
|---|---|---|---|---|---|---|---|---|---|
| Xeros | Xe | 67.5 | 11.0 | 0 | 0 | 100 | 0 | 0 | 0 |
| Agia Forest | Af | 21.3 | 5.5 | 0 | 0 | 100 | 0 | 0 | 0 |
| Stavros | St | 78.9 | 18.9 | 0 | 0 | 42 | 13 | 0.17 | 26 |
| Argaka | Ar | 44.7 | 11.9 | 0 | 0 | 72 | 24 | 0.04 | 0 |
| Pano Gialia | Pg | 15.1 | 4.9 | 0 | 0 | 100 | 0 | 0 | 0 |
| Leivadi | Le | 27.9 | 8.8 | 0 | 4 | 96 | 0 | 0 | 0 |
| Mavros Kremnos | Mk | 5.2 | 2.0 | 0 | 8 | 92 | 0 | 0 | 0 |
| Pyrgos | Py | 38.1 | 12.0 | 0 | 0 | 100 | 0 | 0 | 0 |
| Limnitis | Li | 48.0 | 11.5 | 0 | 0 | 100 | 0 | 0 | 0 |
| Marathasa | Ma | 22.6 | 5.4 | 15 | 65 | 20 | 0 | 0 | 0 |
| Agios Nikolaos | An | 15.7 | 4.8 | 95 | 5 | 0 | 0 | 0 | 0 |
| Platania | Pl | 10.2 | 2.1 | 33 | 67 | 0 | 0 | 0 | 0 |
| Kargiotis | Ka | 64.6 | 13.1 | 30 | 41 | 25 | 3 | 1 | 1 |
| Atsas | At | 32.7 | 15.8 | 0 | 47 | 42 | 8 | 3 | 0 |
| Lagoudera | La | 14.5 | 4.9 | 0 | 12 | 76 | 11 | 0 | 0 |
| Vyzakia | Vy | 81.0 | 15.6 | 0 | 11 | 36 | 38 | 14 | 0 |
| Peristerona | Pe | 78.2 | 13.2 | 1 | 11 | 69 | 20 | 0 | 0 |
| Akaki | Ak | 96.7 | 25.0 | 0 | 2 | 37 | 47 | 11 | 2 |
| Agios Onoufrios | Ao | 14.2 | 11.0 | 0 | 0 | 33 | 57 | 9 | 0 |
| Pedieos | Pd | 29.8 | 16.5 | 0 | 0 | 52 | 35 | 11 | 1 |
| Kotsiatis | Ko | 74.1 | 21.3 | 0 | 1 | 11 | 28 | 59 | 1 |
| Nisou | Ni | 95.6 | 30.3 | 0 | 0 | 9 | 22 | 50 | 18 |

## 3 Data

### 3.1 Streamflow data

For the 22 watersheds, daily discharge data ($m^3 s^{-1}$) from streamflow stations of the Cyprus Water Development Department for the period 1980-2010 were analyzed. In addition, the original continuous hydrograph charts (water levels) of 16 of the 22 streamflow stations, for the Jan-1989 and Nov-1994 events, were scanned and manually digitized through the GetData Graph Digitizer software (http://getdata-graph-digitizer.com). The digitized water levels were interpolated to obtain values precisely every 15 minutes (00.00, 00.15, 00.30, 00.45, 01.00….) and converted to discharge with the appropriate rating curve of the station. The streamflow stations and rating curves are maintained by the Water Development Department through frequent observations. Both interpolation and conversion were carried out by R scripts (https://www.r-project.org/). The 15-minute data were aggregated into hourly discharge values. Both hourly and daily values were used for model performance analysis.

### 3.2 Meteorological data

An hourly gridded dataset with a resolution of $1 \times 1$ km$^2$ was developed using hourly and daily rainfall data from the Cyprus Department of Meteorology stations and the daily gridded rainfall dataset of Camera et al. (2014). Data were extracted for two extreme events, with 42 rain gauges available over the island for Jan 1989 and 37 rain gauges available for Nov 1994. The temporal disaggregation from daily to hourly gridded rainfall was developed through a FORTRAN code based on the method of hourly fractions (Di Luzio et al., 2008), which preserves the original daily values. The main steps of the disaggregation method are:

    a.  The hourly rainfall observations (*ph*) are summed in 24-hour totals (*phs*). The 24-hour period ranges from 8.00 AM of the previous day until 8.00 AM of the attribution day, coherently with the daily gridded dataset.

    b.  The fractions of the hourly rainfall data to the daily total rainfall are calculated as:

$$hfrac = ph/phs. \tag{1}$$

    c.  The nearest gauge to each rainfall gridded dataset cell (*ng*) is found.

    d.  The hourly rainfall at each grid cell (*phc*) is calculated by multiplying each gridded daily (*d*) rainfall value (*pdc*) with the hourly (*h*) fraction (*hfrac*) of the nearest valid gauge (*ng*).

$$phc(h,ng) = pdc(d,c) \cdot hfrac(h,ng). \tag{2}$$

## 4 Modelling setup

### 4.1 WRF-Hydro model description

The WRF-Hydro model is an extension package of the 1-D Noah LSM and Noah-MP LSMs, which are commonly coupled to WRF. In this study, the Noah LSM 2.7.1 version and the WRF-Hydro 3.0 version, as modified by Rummler et al. (2019), were used. WRF-Hydro, in comparison to the traditional 1-D LSM, enhances the physical description and mathematical resolution of surface and near surface hydrologic processes. It includes physics options for quasi 3-D saturated subsurface flow, 1-D or 2-D surface overland flow, 1-D channel routing, lake/reservoir routing, and baseflow processes. WRF-Hydro uses a disaggregation-aggregation procedure to resolve the hydrologic processes at a finer resolution than the LSM. Below, a brief description of the main modeled processes and characteristics is presented. For a detailed description of the model components the reader can refer to Gochis et al. (2015). A schematic representation of the model structure, as used in this study, is presented in Fig. 2.

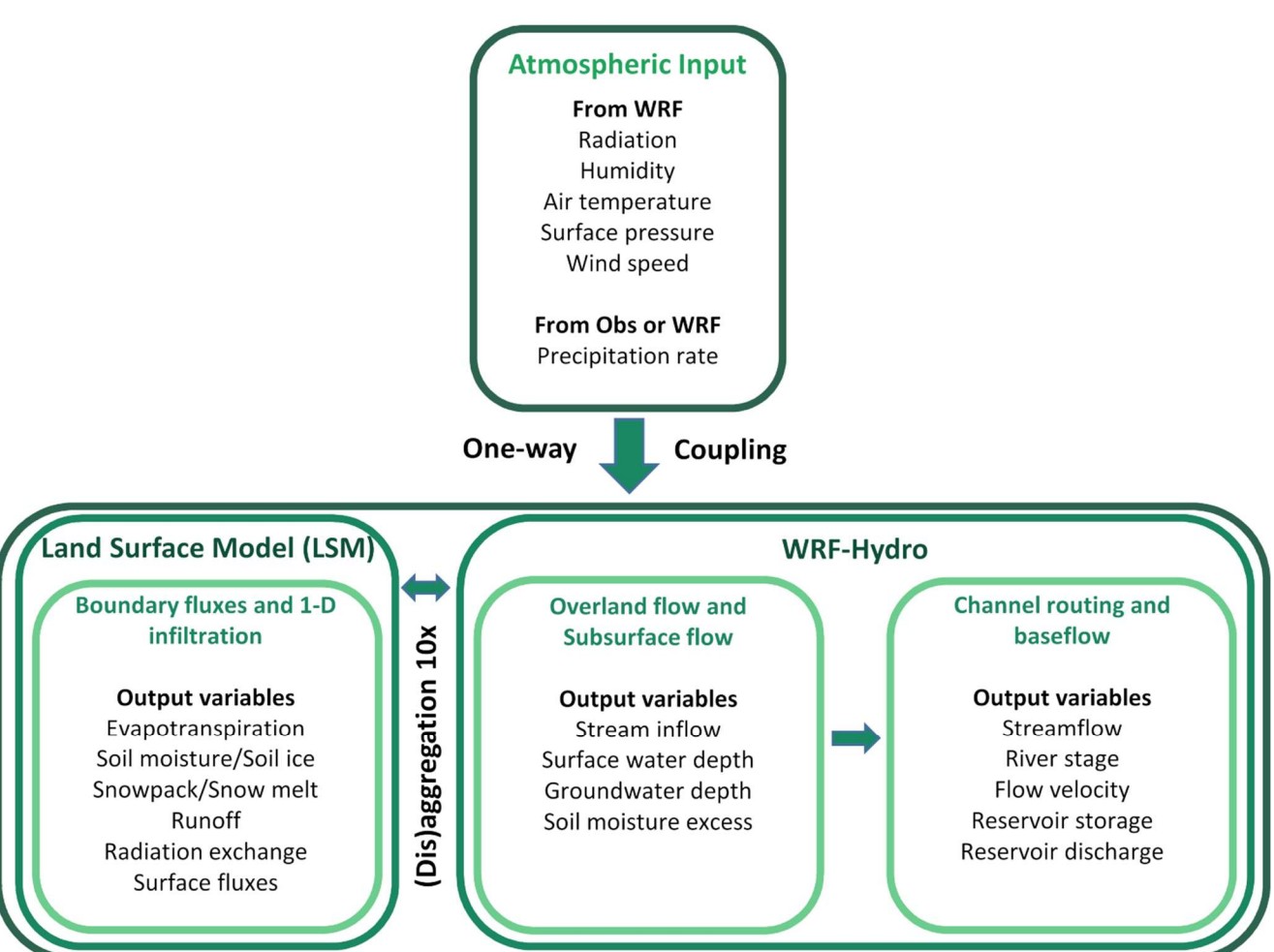

**Figure 2. Schematic illustration of the model structure used in this study, including the coupling between WRF, the Noah Land Surface Model and WRF-Hydro routines (modified after Gochis et al., 2015).**

One of the major advances of WRF-Hydro is the lateral subsurface flow component, which is calculated following the approach

proposed by Wigmosta et al. (1994) and Wigmosta and Lettenmaier (1999). When precipitation reaches the surface, it can either infiltrate or run off. The partitioning between infiltration and runoff is controlled, besides the antecedent soil moisture conditions, by soil properties. In the Noah-LSM, the infiltration capacity (DDT) is defined as a function of the soil moisture deficit (DD) and an exponential scaled adjustment (VAL), which is a function of the parameter KDT. It follows the approach of Schaake et al. (1996), with the difference that KDT is not directly calibrated but is expressed as a function of the saturated

hydraulic conductivity and two scaling coefficients:

$$DDT = DD \cdot VAL, \tag{3}$$

$$VAL = \left(1 - e^{(-KDT \cdot DT)}\right), \tag{4}$$

$$KDT = \frac{REFKDT \cdot K_{sat}}{REFDK}, \tag{5}$$

where DT is the time step duration [day]; $K_{sat}$ [m s$^{-1}$] is the saturated hydraulic conductivity; REFDK is the reference (silty

clay loam) saturated hydraulic conductivity (default 2E-06 m s$^{-1}$); and REFKDT is the infiltration partitioning scaling coefficient, which needs to be calibrated to empirically correct KDT for natural variability. As was demonstrated by previous studies (e.g., Naabil et al., 2017; Verri et al., 2017; Givati et al., 2016; Senatore et al., 2015), the model is sensitive to REFKDT. Once the water enters the soil, it moves vertically, through a four-layer soil column, until it reaches the saturated level and then laterally, according to the local gradient. In case the moisture content at the top of the soil column is larger than its water

holding capacity (saturation), exfiltration occurs. The exfiltration amount is added to the infiltration excess and is routed over the surface. At the bottom of the soil column a vertical flux is calculated, using Richards equation (Richards, 1931). Drainage from the soil column is computed by multiplying the vertical flux with the SLOPE parameter, which can vary between 0-1,

where 0 represents an impermeable boundary between the soil column and the underlying formations. The SLOPE parameter is assigned based on terrain slope classes through a table, however in an implicit way it expresses bedrock properties too (the higher the slope, the higher is the SLOPE coefficient in order to scale the projected map area over which deep drainage occurs). Drained water can be considered a loss or added to streamflow within the channel network through a conceptual baseflow module, if this is activated.

Regarding overland flow, WRF-Hydro allows water to pond on the earth's surface. A water retention depth is defined based on land use and vegetation cover. This parameter can be adjusted through a scaling factor (RTDPT), which can be specified for each model cell and can vary between 1-10 (Yucel et al., 2015). The fraction of ponded water exceeding the retention depth is available to overland flow routing. The routing is performed based on the diffusive wave formulation of Julien et al. (1995) and it can be resolved in both 1-D (Steepest Descent) or 2-D (x-y directions). Overland roughness is defined through the same tables as the retention depth and it can be adjusted through the overland-roughness routing factor (OVRGH), which can vary between 0-1 (Yucel et al., 2015). Overland flow can re-infiltrate, evaporate or enter the channel network.

Water entering the channel network, which the user defines through a Digital Elevation Model, is routed based on a streamflow algorithm that uses an implicit, one-dimensional, variable time stepping diffusive wave formulation. Such formulation is a simplification of the St. Venant equations for shallow water flow. The algorithm does not allow overbank flow and therefore the 2-D modelling of floods (Rummler et al., 2019). Channels are considered trapezoidal in section. Their geometrical properties, including roughness, are defined based on stream order. These model parameters are entered through a table and they can be set by expert knowledge or adjusted during calibration. Along the channel network, reservoirs can be added. Water can flow into reservoirs through the channel network or when overland flow intersects them. Water can flow out of the reservoir through weir overflow and gate-controlled flow. These fluxes are governed by the reservoir parametrization (reservoir area, maximum water level in the reservoir, weir length, gate area, gate elevation, gate aperture coefficient). No exchanges occur between the reservoir, the atmosphere, and the soil column around the reservoir (i.e., evaporation and subsurface lateral flow from the reservoir are not accounted for).

When deep drainage from the soil column is not considered as a loss, WRF-hydro allows two mathematical simple solutions to account for baseflow. For both solutions, baseflow is calculated within sub-watersheds. The first solution consists of a simple pass-through model, meaning that the cumulated deep drainage occurring in a time step is equally redistributed to all channel segments within the sub-watershed. The second solution consists of calculating a baseflow discharge [m$^3$ s$^{-1}$] ($Q_{bf}$) by means of an exponential bucket model, described by the following equation:

$$Q_{bf} = C \cdot \left( e^{a \cdot \frac{Z}{Z_{max}}} - 1 \right), \tag{6}$$

where $C$ is the bucket coefficient [m$^3$ s$^{-1}$], $a$ is the bucket model exponent [-], $Z_{max}$ is the maximum bucket level [m], and $Z$ [m] is the bucket level at a certain time step. The user defines the $C$, $a$ and $Z_{max}$ parameters for each sub-watershed, together with a $Z_{ini}$ [m] parameter to initialize the water storage in the bucket groundwater reservoir. At each time step the $Z$ value is updated first adding the deep drainage contribution (Perc) and subsequently subtracting $Q_{bf}$:

$$Z_t = Z_{t-1} + \sum_{n=1}^{n=ncells} Perc_n - \frac{Q_{bf} \cdot DT \cdot 3600}{A} \tag{7}$$

where A is the area of the sub-watershed [m$^2$], DT the model time step [day], n is the index for the sub-watershed cells, and ncells represents the number of cells of the sub-watershed. Similar to the first solution, $Q_{bf}$ is equally redistributed to channel segments. If Z equals or exceeds $Z_{max}$, all deep drainage is transferred to the channel network.

## 4.2 WRF-Hydro model parameterization

The Noah LSM was parameterized over a 1 × 1 km$^2$ grid, while WRF-Hydro was run over a 100 × 100 m$^2$ grid. All simulations were performed in uncoupled mode, resolving the steepest descend formulation of the overland flow routine, with channel flow, baseflow and reservoir routines activated.

To run WRF-Hydro in uncoupled mode, the meteorological forcing needed are precipitation rate [mm s$^{-1}$], downward shortwave and longwave radiation [W m$^{-2}$], specific humidity [kg kg$^{-1}$], air temperature [K], surface pressure [Pa], near surface wind components [m s$^{-1}$]. For the calibration and validation runs, all variables except precipitation were taken from the WRF ERA-Interim downscaling experiments presented in Zittis et al. (2017). These simulations incorporated the Grell-Freitas Ensemble Convection and the Ferrier Microphysics parameterization schemes, which were found to outperform the other tested configurations for the selected events. For precipitation, hourly observed gridded data were used (see Section 3.2 – Meteorological Data). For the simulation runs with WRF-modelled rainfall, all variables including precipitation were taken from the WRF experiments (Zittis et al., 2017). To derive soil moisture initial conditions, 15-day WRF spin-up runs were performed for both events. For Jan 89, the 15-day rainfall during spin-up was 99 mm and average soil moisture at the end of the simulation was 0.32 m$^3$ m$^{-3}$. The Nov-1994 event followed the dry summer and only a few scattered rain days occurred between the end of October and the beginning of November. The 15-day rainfall during spin-up was 18.4 mm and average soil moisture at the end of the simulation was 0.26 m$^3$ m$^{-3}$. Experimental data (Camera et al., 2018) show that in these conditions soil moisture for a gravelly sandy loam at 1300 m a.s.l. in the Troodos Mountains can vary between 0.10 and 0.15 m$^3$ m$^{-3}$. Therefore, the WRF-derived initial soil moisture values for November were halved.

Land use and vegetation cover data were derived from the MODIS dataset through the WRF Pre-Processing System. According to the MODIS dataset, the Troodos Mountains has a uniform clay loam texture. However, field observations at higher elevation in the mountains, where the predominant lithologies consist of gabbro and ultramafic rocks, showed a gravelly sandy loam texture (Djuma et al., 2020; Camera et al., 2018; Cyprus Geological Survey Department, 1995). In addition, it is known that the Troodos gabbro is very weathered and therefore permeable (Christofi et al., 2020). Therefore, a sandy loam soil type was assigned to these areas. The related properties were attributed through the default table values implemented in WRF-Hydro (see Gochis et al., 2015). The hydrologic input layers (latitude, longitude, topography, flow direction, channel grid, lake grid, stream order, watersheds) were all calculated in ArcGIS® 10.2.2 starting from a 25 × 25 m$^2$ Digital Elevation Model (see Camera et al., 2017), resampled on the 100 × 100 m$^2$ grid, and the known locations of stream gauges and lakes. For the channel grid, a flow accumulation threshold of 250 cells (2.5 km$^2$) was adopted.

For the definition of the deep drainage related parameter, two approaches were tested. First, nine slope terrain classes were derived following Silver et al. (2017). In the second case, for cells where the bedrock consists of gabbro or ultramafic rocks (Cyprus Geological Survey Department, 1995), the slope terrain class (3) that maximizes drainage (representing a highly fractured system) was assigned. In both cases, for each slope terrain class, the related default SLOPE value listed in the WRF-hydro general parameters table was given. These changes in soil type and deep drainage based on geology affected mainly watersheds Ma, An, Pl, Ka, and At, where 70% or more of the surface bedrock is made up of gabbro and ultramafic rocks (Table 1).

Other general parameters are REFKDT and soil depth (SD), which were calibrated. REFDK was left to its default value (2.00E-6 m s$^{-1}$). The WRF-Hydro parameter OVRGH was tested and values were assigned based on the sensitivity analysis, whereas RTDPT was kept constant all over the study area and a value of 1, consistent with a steep mountainous terrain, was assigned. Channel geometrical parameters were attributed based on the study area knowledge of the authors (Table 2). The initial channel water depth was set to the default value for dry conditions. Six reservoirs were characterized in the model setup (Table 3) according to data from the Cyprus Water Development Department (2009). At all reservoirs, outflow occurs for overflow only; the structures do not have a gate. Vyzakia reservoir was completed in early 1994, therefore it was not included in the Jan-1989 simulation.

Regarding baseflow, the parameter $C$ was set equal to the long-term baseflow index, calculated from the 1980-2010 data series with the program PART (Rutledge, 1988). The initial level of the conceptual reservoir ($Z_{ini}$) was set as a fraction of the maximum level ($Z_{max}$), based on the saturation degree of the deepest soil layer at the end of the 15-day WRF spin-up period. The exponent $a$ and $Z_{max}$ were adjusted during calibration.

## 4.3 WRF-Hydro sensitivity analysis

A sensitivity analysis of the LSM parameters REFKDT, SLOPE, and soil depth (SD), which have been identified as sensitive parameters in previous studies (e.g., Fersch et al., 2019; Senatore et al., 2015), was performed for the Jan-1989 event. In addition, sensitivity runs for the OVRGH parameter and the saturated hydraulic conductivity ($K_S$) were performed, too. For these simulations, the baseflow routine was switched off. A reference scenario was set, with REFKDT and OVRGH equal to 1, SD equal to 1.0 m, $K_S$ equal to 2.45E-6 m s$^{-1}$ (value attributed to clay loam soils in the soil parameter table), and the deep drainage parameter (SLOPE) assigned based on terrain slope, as in Silver et al. (2017). Parameters were changed one at a time. Eight values were tested for REFKDT (0.3, 0.5, 3.0, 5.0, 8.0, 10.0, 100.0, 1000.0), two for SD (0.5 and 2.0 m), two for OVRGH (0.1, 0.5), three for $K_S$ (3.38E-6 m s$^{-1}$ as for loam, 5.23E-6 m s$^{-1}$ as for sandy loam, 1.41E-5 m s$^{-1}$ as for loamy sand), and a different set of SLOPE values was assigned based on terrain slope and geology. Also, to demonstrate the equifinality of calibrating REFDK and REFKDT, as suggested by eq. 5, two extra runs were performed for REFDK values of 4.00E-6 m s$^{-1}$ and 6.67E-7 m s$^{-1}$. The relative sensitivity ($S$) was computed according to the following formula:

$$S = -\frac{(Vtot_i - Vtot_{ref})}{Vtot_{ref}}, \tag{8}$$

where $Vtot$ is the total volume discharged during the simulation period, $ref$ refers to the reference scenario, and $i$ to the perturbed value.

**Table 2. WRF-Hydro channel parameter values used in this study (*Bw* is the channel bottom width, *HLINK* is the initial depth of water in the channel, *ChSSlp* is the channel side slope, and *MannN* is the Manning's roughness coefficient).**

| Stream Order | *Bw* [m] | *HLINK* [m] | *ChSSlp* [-] | *MannN* [-] |
|---|---|---|---|---|
| 1 | 1.5 | 0.02 | 3.00 | 0.14 |
| 2 | 3.0 | 0.02 | 1.00 | 0.12 |
| 3 | 5.0 | 0.02 | 0.50 | 0.09 |
| 4 | 10.0 | 0.03 | 0.18 | 0.09 |

**Table 3. Characteristics of the reservoirs included in the WRF-Hydro simulations; Long and Lat are longitude and latitude, respectively.**

| Watershed | Reservoir Name | Long [deg] | Lat [deg] | Reservoir Area [m$^2$] | Reservoir max elevation [m a.s.l.] | Reservoir ave elevation [m a.s.l.] | Weir length [m] |
|---|---|---|---|---|---|---|---|
| Vyzakia | Xyliatos | 33.038 | 35.006 | 80000 | 537.5 | 529.9 | 15.0 |
| Vyzakia | Vyzakia | 33.029 | 33.029 | 160000 | 353.8 | 319.0 | 6.0 |
| Akaki | Palaichori | 33.130 | 34.928 | 110000 | 719.6 | 704.5 | 9.8 |
| Akaki | Kalochorio | 33.155 | 34.981 | 13000 | 533.5 | 528.5 | 22.5 |
| Kotsiatis | Lythrodontas-1 | 33.274 | 34.944 | 10000 | 460.3 | 455.3 | 19.0 |
| Kotsiatis | Lythrodontas-2 | 33.288 | 34.949 | 15000 | 422.5 | 413.5 | 33.8 |

## 4.4 WRF-Hydro calibration and validation with observed precipitation

Calibration runs were evaluated for each watershed against Jan 1989 daily observed streamflow, based on five performance indices. The selected set of indices contains both absolute error and goodness-of-fit measures, as suggested by Legates and McCabe (1999). They are percent bias *(PBIAS),* Mean Absolute Error (*MAE*), Nash-Sutcliffe Efficiency (*NSE* - Nash and Sutcliffe, 1970), modified Nash-Sutcliffe Efficiency (*mNSE*, Krause et al., 2005), and Kling-Gupta Efficiency (*KGE*, Kling et al., 2012).

Soil Depth is constant throughout the domain, therefore it was fixed at the value that returned the best performance indices in the majority of the watersheds, following an evaluation of the sensitivity analysis runs. Similarly, SLOPE parameters were assigned using the slope terrain class map allowing the best performance during sensitivity. REFKDT and OVRGH were initialized, in each watershed, based on the evaluation of the sensitivity runs through performance indices, as for SD. For the baseflow bucket routine, initial values of $\alpha$ and $Z_{max}$ were set to the default. Next, the initialized parameters were fine-tuned based on a trial and error procedure for all watersheds. Modifications were applied to a single parameter at the time and if

changes could not improve the model performance according to three indices out of five after five attempts, the parameters were retained. Commonly applied changes were ±1 for REFKDT, ±0.1 for OVRGH, ±0.5 for $\alpha$, and ±10% of the actual value for $Z_{max}$. Smaller (larger) changes were applied only in watersheds where the response of streamflow was (not) particularly sensitive to specific parameters. The parameterization of $Z_{max}$ was aimed at filling the reservoir after the rainfall peak, between 10 January at midnight and 11 January at noon, to simulate the observed recession of the hydrograph. For those watersheds that highly overestimated the baseflow due to spilling out of the groundwater reservoir, $Z_{max}$ was further increased. A good fit between observed and simulated flow before the peak was the target for the calibration of the exponent α. The calibrated model was subsequently applied to the Nov-1994 event for validation. The same five model performance indices were used for the evaluation.

### 4.5 WRF-Hydro simulations with WRF-modelled precipitation

The WRF-modelled precipitation (Zittis et al., 2017) was averaged over each of the 22 watersheds and the daily values were compared to observed data by means of *BIAS*, *MAE* and *NSE*. To evaluate how deviations from the observed rainfall pattern affected the hydrologic model performance in these small mountain watersheds, the calibrated version of WRF-Hydro model was run with the WRF-modelled hourly precipitation forcing. Modelled streamflow was evaluated with observed data, similar as in the calibration phase.

### 4.6 WRF-Hydro evaluation with observed and modeled precipitation at hourly scale

For watersheds presenting daily *NSE* equal to or larger than 0.50 for both the calibration and the validation event, model performance was also investigated at hourly resolution. The *NSE*, *KGE* and *MAE* were computed for the hourly streamflow values simulated with both observed and modeled precipitation.

## 5 Results and discussion

### 5.1 Sensitivity analysis

The results of the sensitivity analysis are presented in Fig. 3 as boxplots. Each boxplot represents the sensitivity of the modelled total discharge volume, over the 22 watersheds, for the perturbation applied, in comparison to the reference simulation. The boxplots show that in the suggested calibration range (0.5-5.0, Gochis et al., 2015) REFKDT is very sensitive. Although the sensitivity decreases for REFKDT values larger than 5.0, variations in the discharged volume can be observed up to REFKDT values equal to 100.0. Further increases in REFKDT (see REFKDT 1000.0) do not cause any variations in discharge, suggesting that the model already infiltrates at its maximum capacity. The variability over the watersheds is related to local conditions (e.g., soil moisture distribution, area, topography, type of vegetation). Precipitation, which is not homogeneous throughout the study area, can play a role in causing different responses as well.

The two simulations ran with REFDK values of 4.00E-06 m s$^{-1}$ and 6.67E-7 m s$^{-1}$ returned discharged volumes equal to those obtained with REFKDT values of 0.5 and 3.0, respectively. These results confirm the equifinality of the two parameters and make it clear that REFDK calibration should be avoided. As shown in Eq (3-5), REFDK automatically adjusts the infiltration capacity for the effect of soil texture, whereas any other effects on the partitioning of rainfall into surface runoff and infiltration can and should be calibrated through REFKDT.

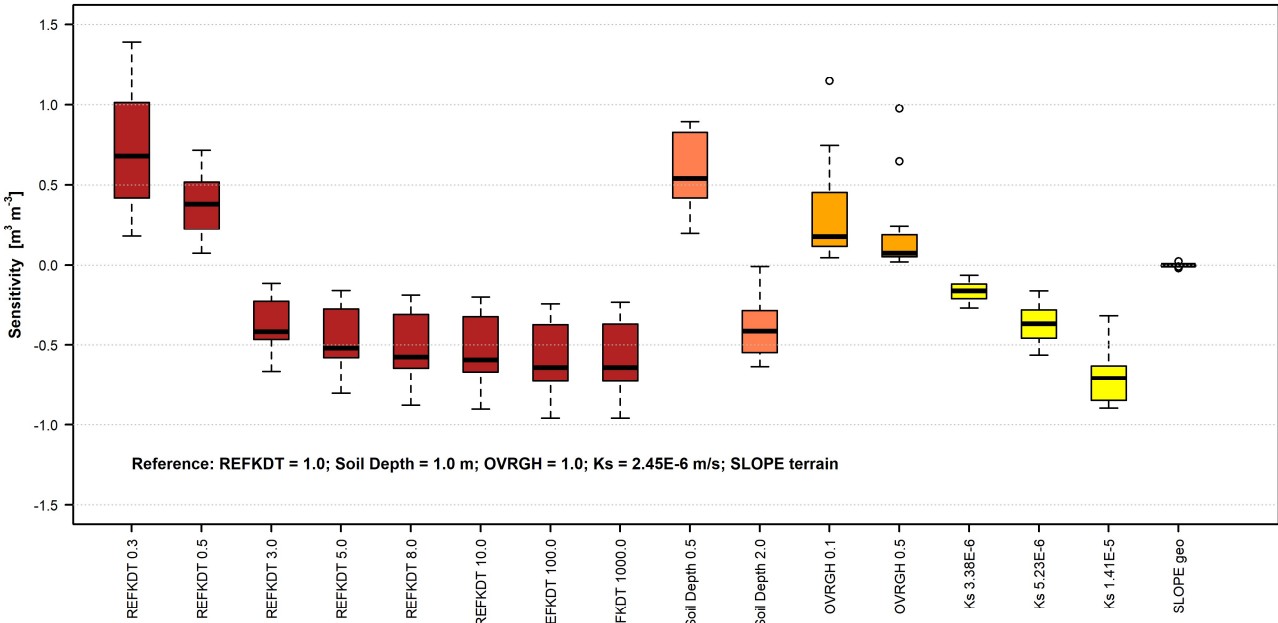

**Figure 3. Boxplots of the sensitivity of the modelled streamflow to perturbations (x-axis) in REFKDT (infiltration partitioning scaling coefficient), Soil Depth, OVRGH (overland roughness factor), $K_S$ (saturated hydraulic conductivity), and SLOPE geo (deep drainage parameter defined based on slope terrain and geology) relative to a defined reference scenario (SLOPE terrain represents the slope parameter defined based on slope terrain only, as in Silver et al., 2017).**

The sensitivity analysis shows also an important role played by Soil Depth. Especially in mountainous areas, soils are usually thin. This limited soil thickness affects the total amount of water retained by the soil, favoring a partitioning of the available water between infiltration and surface runoff towards the latter. Similar observations are reported by Fersch et al. (2019), while commenting the offset between modelled and observed soil moisture content in mountainous catchments in Bavaria (Germany). To overcome the issue, in other land surface models (e.g., Brunke et al., 2016) variable soil thickness has been implemented and tested.

Regarding OVRGH, results show that it has a slight control on the total volume discharge, as also presented in Yucel et al. (2015), while it has almost no effect on delaying the peak (Fig. 4). More sensitive than OVRGH is Ks, suggesting a possible important impact of the soil type and property definitions on the model output. Senatore et al. (2015) presented one of the few WRF-Hydro studies that calibrated a hydraulic conductivity related parameter, although they focused on the saturated soil lateral conductivity. SLOPE appeared to have a low sensitivity, although in the mountain watersheds, where it changed, a small reduction in the total discharged volume was observed.

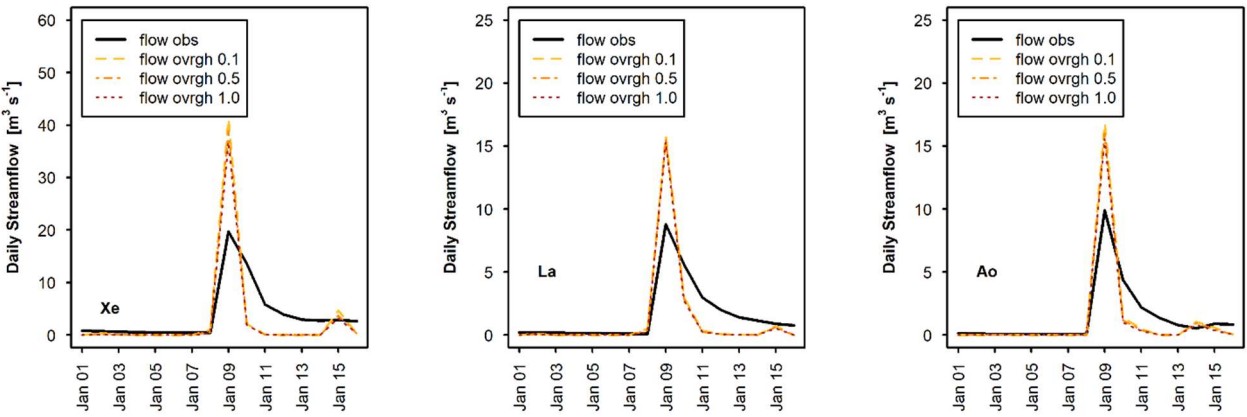

**Figure 4. Hydrographs obtained at three different watersheds for OVRGH values of 0.1 (flow ovrgh 0.1), 0.5 (flow ovrgh 0.5), and 1.0 (flow ovrgh 1.0), in comparison to observed flow (flow obs). For watershed short names refer to Table 1.**

## 5.2 WRF-Hydro calibration and validation with observed precipitation

The calibrated parameters are listed in Table 4. Soil depth was set equal to 1 m for all watersheds, because it was the value returning the best performance indices (Fig. 4) in 16 out of 22 catchments (average *NSE* improvement equal to 0.14). SLOPE attributed based on both terrain slope and geology resulted in slightly better performance indices in the mountain watersheds than SLOPE attributed through terrain slope only. Therefore, it was selected for the final parameterization. Also, for all watersheds OVRGH was set equal to 1 because it was the value returning the best performance indices in 19 out of 22 watersheds. Furthermore, considering that OVRGH effects total discharge volume and not hydrograph shape, its calibration would have been equifinal to REFKDT. Twelve watersheds have a REFKDT coefficient larger than 5.0, which is outside the 0.5-5.0 range suggested by Gochis et al. (2015), but none has a REFKDT lower than 0.5. The hydrographs of all watersheds are shown in the supplementary material. Fig. S1 and Fig. S2 show hydrographs, including the baseflow component, related to responses to observed rainfall for the Jan-1989 event and the Nov-1994 event, respectively.

The parameterization of watersheds Ma, An, Pl, Ka, and At is peculiar. These watersheds are mainly characterized by sandy loam texture (i.e., higher Ks than the other watersheds), maximum deep drainage obtained by using the SLOPE parameters based on slope terrain and geology, very high REFKDT values, and very large groundwater storage. However, poor model fit indices (for some watersheds even negative) were obtained for the calibration period (Fig. 5). Conversely, the same watersheds show positive *NSE* values and negative *PBIAS* (i.e., slight underestimation of the peak discharge), for the validation event. Overestimation of runoff in Jan 1989 could have been related to the modeling of snow and snowmelt in the LSM. Both observed and modeled temperature values for the upstream areas of these watersheds showed negative values, indicating that part of the precipitation was snow. In Fig. 6, the comparison between the observed and simulated daily hydrographs for the Jan-1989 event is shown. The subdued response of the streamflow to the extreme precipitation is clear for watershed Pl, which is considered representative of the behavior of all five watersheds mentioned above, and it is clear that the simulated hydrograph overestimates the observed peak flow of the event. Different bottom boundary conditions and snow processes modelling, as those implemented in the Noah Multi-Physics LSM, could improve the simulation results.

**Table 4. Calibrated parameters (REFKDT, infiltration partitioning scaling coefficient; *C*, baseflow bucket coefficient; *α*, bucket exponent; *Z$_{max}$*, maximum bucket level) for the 22 watersheds with their maximum (MaxQ) and average (AveQ) discharges for the two analyzed events; for watershed short names refer to Table 1.**

| Watershed short name | Max Q89 [m$^3$ s$^{-1}$] | Ave Q89 [m$^3$ s$^{-1}$] | Max Q94 [m$^3$ s$^{-1}$] | Ave Q94 [m$^3$ s$^{-1}$] | REFKDT [-] | C [m$^3$ s$^{-1}$] | α [-] | Z$_{max}$ [m] |
|---|---|---|---|---|---|---|---|---|
| Xe | 19.7 | 3.7 | 8.2 | 1.1 | 5.0 | 0.30 | 2.0 | 30.0 |
| Af | 4.1 | 1.0 | 1.1 | 0.2 | 50.0 | 0.09 | 0.7 | 150.0 |
| St | 12.5 | 2.7 | 4.1 | 0.6 | 2.5 | 0.20 | 2.4 | 100.0 |
| Ar | 3.2 | 1.0 | 1.5 | 0.2 | 12.0 | 0.08 | 2.6 | 70.0 |
| Pg | 1.1 | 0.3 | 0.3 | 0.1 | 7.0 | 0.04 | 1.1 | 3.3 |
| Le | 3.3 | 0.9 | 1.4 | 0.3 | 1.8 | 0.07 | 2.0 | 1.5 |
| Mk | 0.3 | 0.1 | 0.1 | 0.0 | 8.0 | 0.01 | 3.2 | 20.0 |
| Py | 4.1 | 1.4 | 2.2 | 0.4 | 5.0 | 0.15 | 1.4 | 200.0 |
| Li | 12.3 | 3.0 | 5.4 | 1.0 | 7.0 | 0.27 | 1.2 | 72.0 |
| Ma | 3.7 | 1.2 | 2.9 | 0.6 | 50.0 | 0.19 | 1.6 | 600.0 |
| An | 1.8 | 0.7 | 3.3 | 0.8 | 50.0 | 0.24 | 1.6 | 600.0 |
| Pl | 1.3 | 0.4 | 1.9 | 0.3 | 50.0 | 0.05 | 2.1 | 600.0 |
| Ka | 9.2 | 2.6 | 10.5 | 1.9 | 50.0 | 0.30 | 1.2 | 500.0 |
| At | 2.9 | 1.0 | 1.9 | 0.5 | 10.0 | 0.04 | 2.1 | 220.0 |
| La | 8.8 | 1.6 | 6.4 | 0.9 | 6.0 | 0.05 | 2.3 | 53.0 |
| Vy | 15.9 | 3.5 | 12.0 | 2.4 | 4.0 | 0.09 | 2.6 | 50.0 |
| Pe | 58.0 | 7.5 | 35.0 | 5.9 | 1.0 | 0.29 | 2.5 | 8.0 |
| Ak | 49.0 | 7.7 | 28.0 | 5.7 | 0.8 | 0.20 | 1.4 | 5.0 |
| Ao | 9.9 | 1.4 | 5.3 | 1.1 | 3.0 | 0.03 | 2.1 | 10.0 |
| Pd | 26.0 | 3.2 | 8.9 | 1.9 | 2.0 | 0.07 | 2.2 | 10.0 |
| Ko | 18.0 | 3.6 | 15.0 | 3.0 | 5.0 | 0.05 | 3.2 | 2.4 |
| Ni | 18.2 | 4.1 | 16.5 | 3.0 | 7.0 | 0.05 | 3.2 | 3.0 |

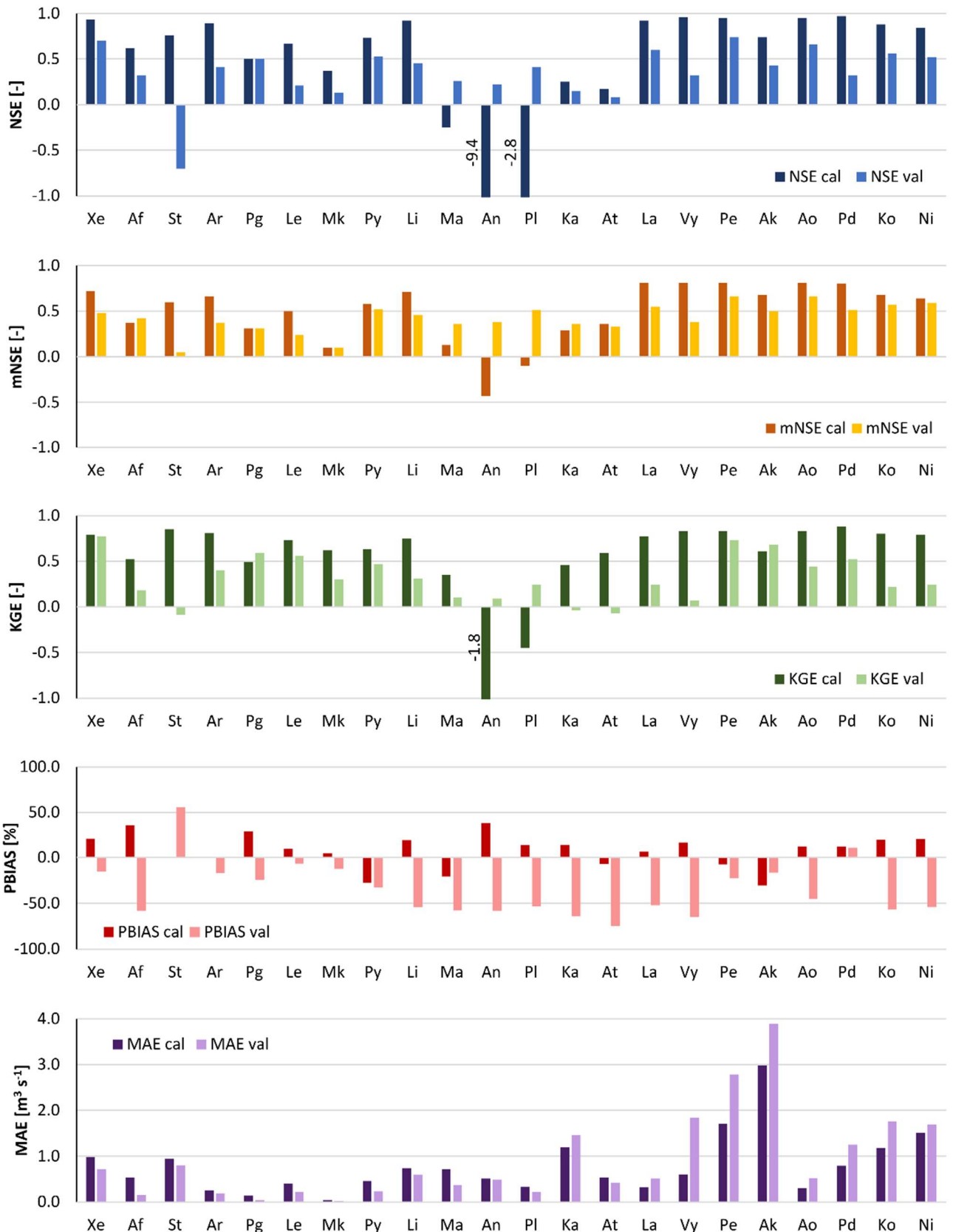

**Figure 5. Performance indices (NSE, Nash-Sutcliffe Efficiency; mNSE, modified Nash-Sutcliffe Efficiency; KGE, Kling-Gupta Efficiency; BIAS; MAE, Mean Absolute Error) calculated on daily streamflow resulting from observed rainfall for the 22 watersheds using the calibrated set of parameters for both Jan 1989 (cal) and Nov 1994 (val). For watershed short names refer to Table 1.**

Overall, in all other watersheds the model behaves satisfactorily, with goodness-of-fit scores (*NSE*, *mNSE* and *KGE*, Fig. 5) usually higher than 0.5 for the calibration run and larger than 0.0 for the validation event. Exceptions are watershed Mk for the calibration run and watershed St for the validation run. Looking at the hydrographs (Fig. 6 and Fig. 7), it is observed that

Mk presents a very low discharge due to its limited area (Table 4). Therefore, small biases between observed and modelled streamflow produce poor goodness-of-fit indices. Also, Mk is the only watershed showing higher rainfall and flow peaks towards the end of the Jan-1989 event rather than in the middle. The model slightly underestimates the flow peak occurred on January 9th and overestimates the flow at the end of the simulation period. For St, the model reacts sharply to precipitation input, simulating well the flow peak occurred on January 9th but overestimating the flow at end of the simulation period of the Jan-1989 event and above all the peak of the Nov-1994 event, therefore affecting the performance scores.

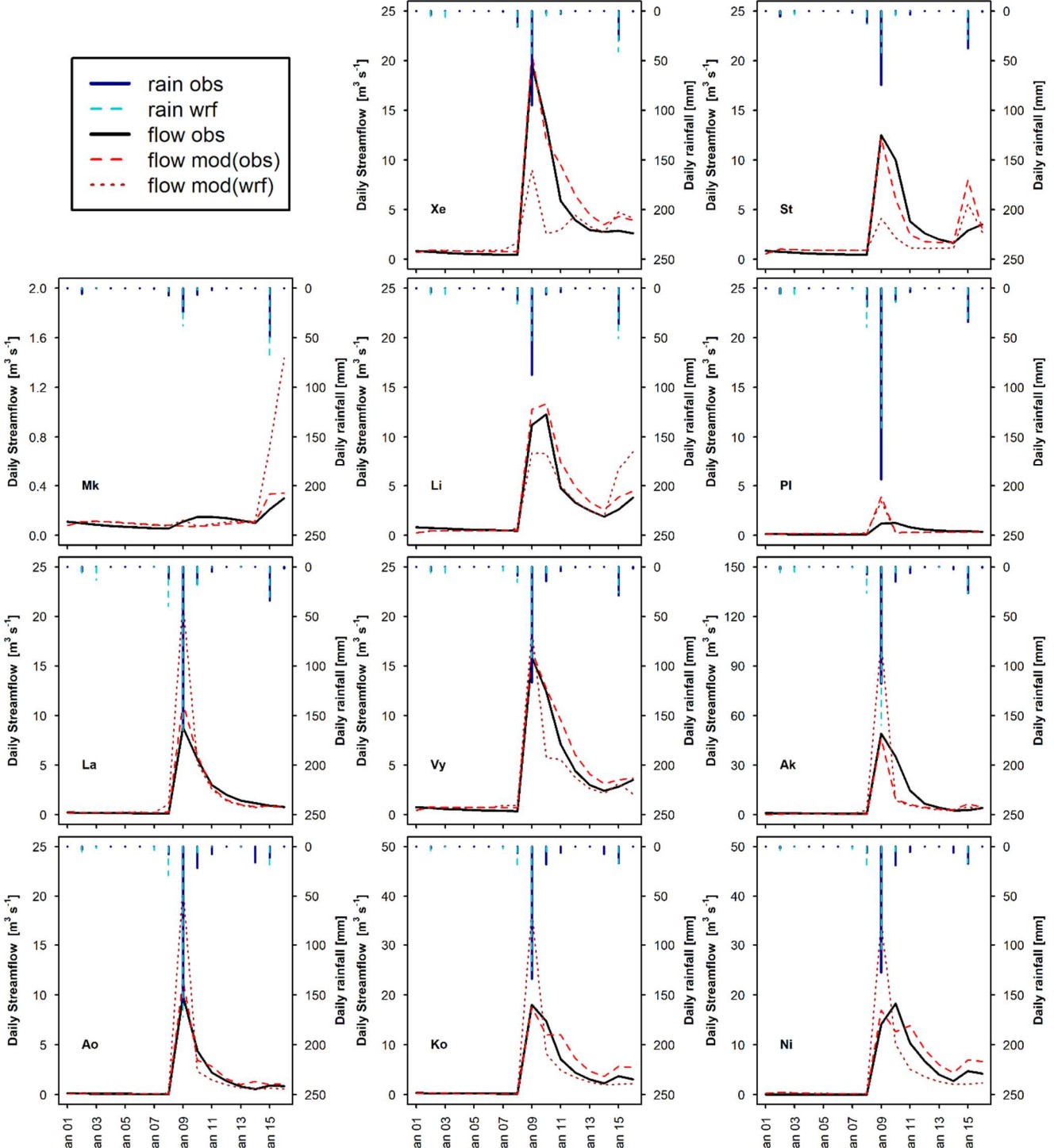

**Figure 6. Observed daily hydrographs (flow obs) and hydrographs obtained with the calibrated WRF-Hydro model (flow mod) forced with observed rainfall (rain obs) and with WRF modelled rainfall (rain wrf) for the Jan-1989 calibration event, for 11 representative watersheds (see Table 1 for watershed short names).**

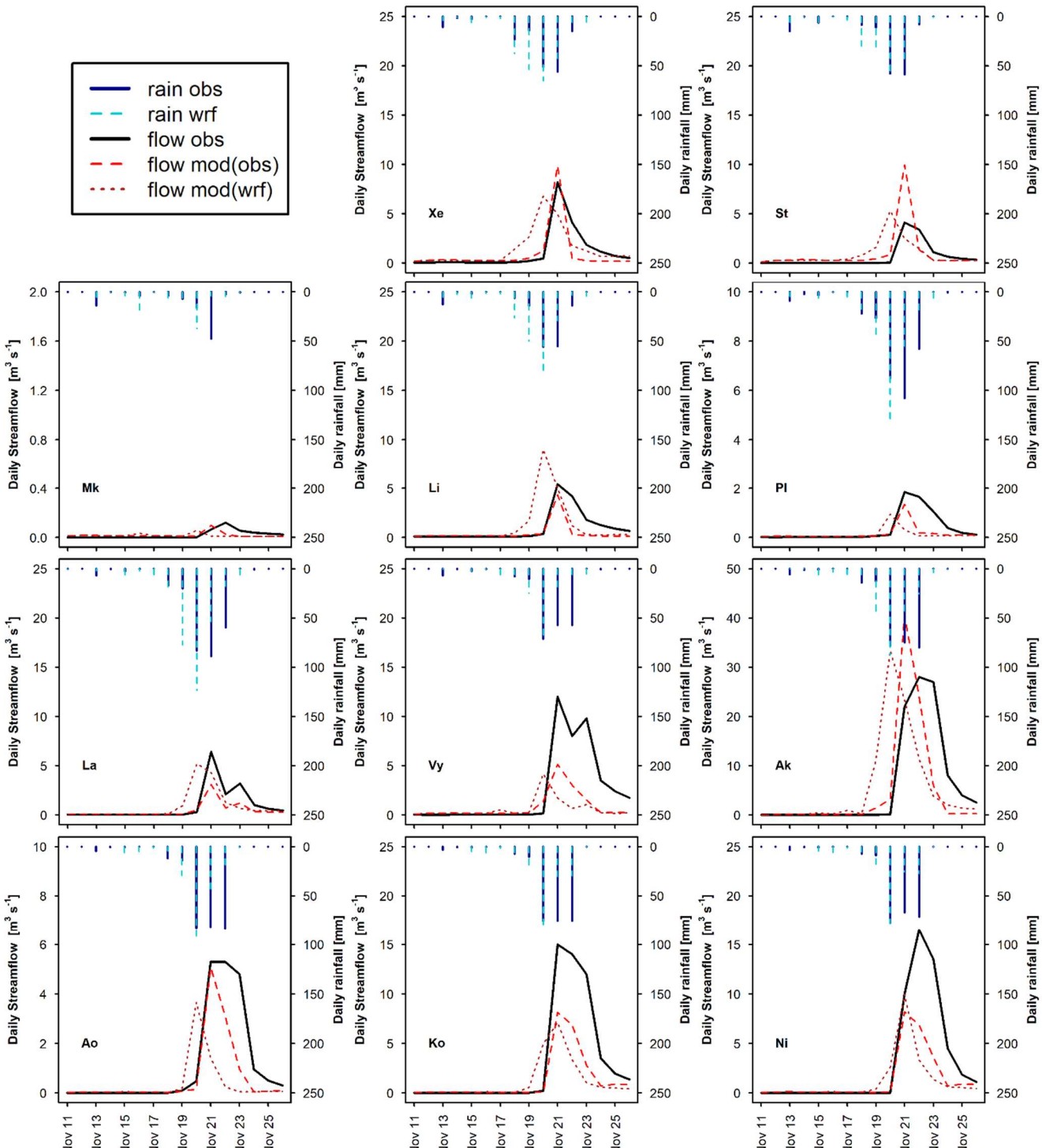

**Figure 7. Observed daily hydrographs (flow obs) and hydrographs obtained with the calibrated WRF-Hydro model (flow mod) forced with observed rainfall (rain obs) and with WRF modelled rainfall (rain wrf) for the Nov-1994 validation event, for 11 representative watersheds (see Table 1 for watershed short names).**

In the eastern part of the modelling domain (La to Ni), for the calibration event both initial baseflow and the discharge peak are well modelled in all watersheds (Fig. 6). Differences between observed and simulated hydrographs can be observed in the post-peak, for watersheds Ak, Pe (Fig. S1), Ko and Ni. Ak and Pe present a very high peak flow (> 50 m³ s⁻¹) and an underestimation of the receding limb of the hydrograph in the following days, which causes the negative *PBIAS* and high *MAE* values visible in Fig 5. In the case of Ko and Ni, the receding limb shows a little overestimation. For the validation event (Fig.

7), the peak is well simulated in Pe and Ao, slightly overestimated in Ak and Pd, underestimated in La, Vy, Ko, and Ni (Pe and Pd, Fig. S2). In the post peak phase, the simulated hydrographs show negative biases in comparison to the observed ones in all watersheds. As it is visible in Fig S1 and Fig S2, flow in the receding limb of the hydrograph is mainly made up of baseflow. For Jan-1989 event, in all these watersheds the groundwater reservoir is filled up on January 10th and baseflow

consists of the water spilling out from it. This water volume, redistributed along the channel network, is generally able to

410 reproduce the hydrograph shape, except in Ak. In Nov 1994, no groundwater spilling is observed during the simulation and the receding limb is underestimated. Therefore, this could be partly due to a non-perfect reproduction of the model initial conditions and partly related to an underestimation of interflow and baseflow.

## 5.3 WRF-Hydro simulations with modeled precipitation

Figure 8 presents the performance indices of the WRF-modelled rainfall. Fig. S3 and Fig. S4 (in the supplementary material)

show hydrographs, including the baseflow component, related to responses to modelled rainfall for all watersheds for the Jan-1989 event and the Nov-1994 event, respectively. The modelled rainfall is generally closer to observations for the Jan-1989 event than for the Nov-1994 event, as testified by the higher *NSE* (except for Le) and lower *MAE* values (Fig. 8). As can be seen in Fig. 6, the Jan-1989 event appears as a single day of intense precipitation, followed by a few scattered low rainfall days that can show a moderate intensity towards the end of the simulation period. During Jan-1989, WRF-modelled rainfall is

usually able to fit the observed daily precipitation trend over all watersheds, with slight variations in the calculated daily amounts as suggested by the generally low bias (Fig. 8). In percentage, over the 22 watersheds rainfall *PBIAS* varies between -35% and 53%, with an average of absolute values equal to 17%. Average *NSE* and *MAE* of the WRF-modelled rainfall are 0.83 and 4.5 mm d$^{-1}$, respectively.

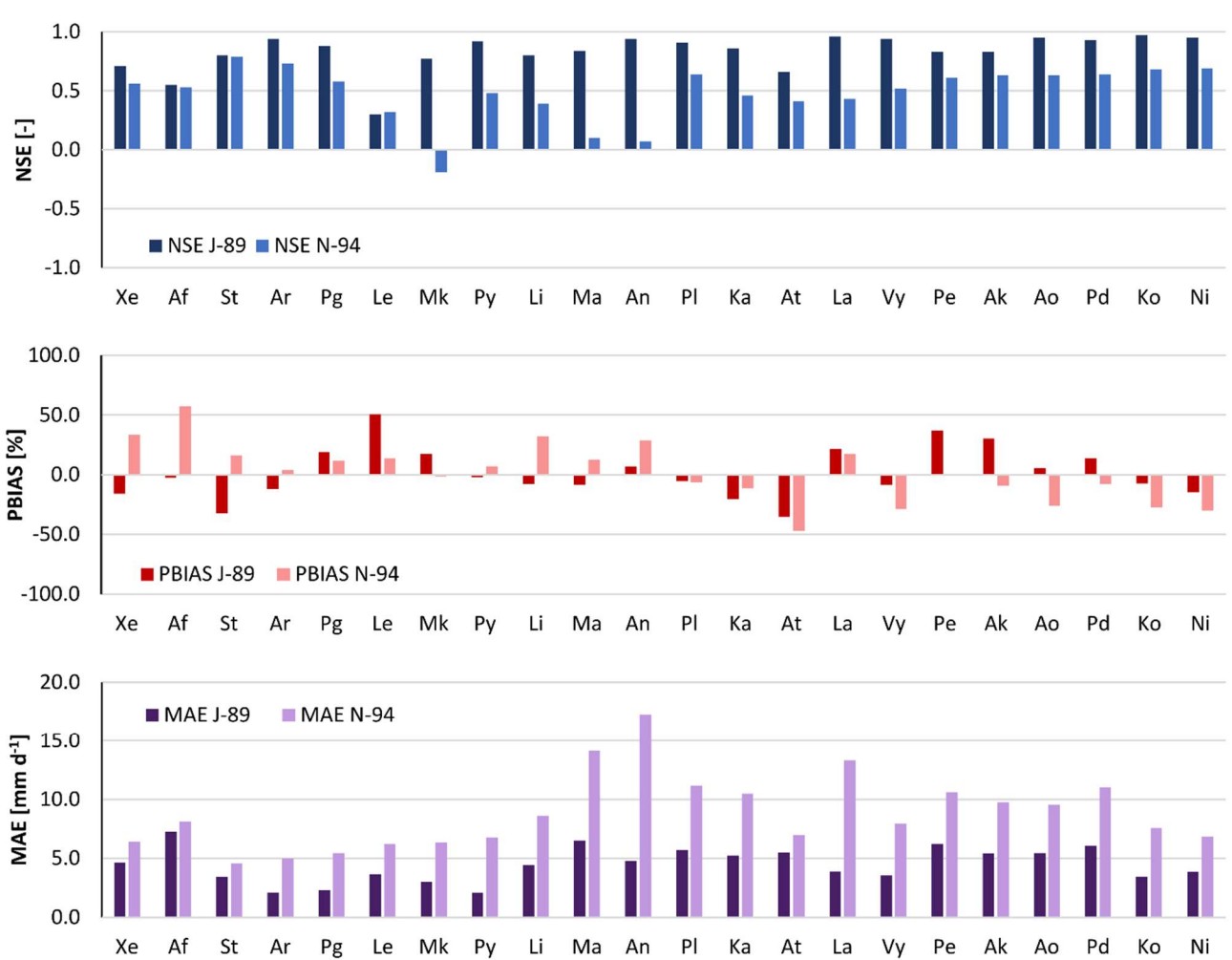

**Figure 8. Performance indices (NSE, Nash-Sutcliffe Efficiency; BIAS; MAE, Mean Absolute Error) of daily WRF-modelled rainfall over the 22 watersheds both Jan 1989 (J-89) and Nov 1994 (N-94) events. For watershed short names refer to Table 1.**

Figure 7 shows that the Nov-1994 event is constituted of two days of moderately low precipitation, followed by three days of intense precipitation. The simulated event shows higher rainfall amounts in the preceding days and a loss of intensity after the first of the three high precipitation days. Over the 22 watersheds, average *NSE*, absolute *PBIAS,* and *MAE* are 0.48, 20%, and 8.9 mm d$^{-1}$, respectively.

The modelled rainfall in Jan 1989 results in hydrograph shapes similar to the observed ones but still in goodness-of-fit indices that are often negative. With observed rainfall forcing, the simulated daily hydrograph returned negative *NSE, mNSE and KGE* values (Fig. 5) in three, two and two watersheds, respectively. With WRF-modelled rainfall forcing the number of watersheds with negative indices (Fig. 9) increases up to twelve, six, and nine, respectively. Moving from observed to WRF-modelled rainfall, both streamflow *NSE* and *MAE* indicate a loss in model performance in all watersheds except three (Ma, Pl, Ka), which are those characterized by very negative goodness-of-fit indices in the calibration run. The average streamflow *MAE* almost doubled, and ranged between 0.09 m$^3$ s$^{-1}$ in Mk and 3.89 m$^3$ s$^{-1}$ in Pe. The absolute value of flow *PBIAS* decreased in seven watersheds (Af, Li, Pl, Vy, Ak, Ko, Ni) but on average increased by 21.5% (96.6% in Pg and 120.3% in Le).

Regarding streamflow for Nov-1994 event, the peak discharge is simulated to occur one day earlier than observed in most watersheds. This caused negative streamflow performance indices in eighteen watersheds for *NSE*, in eight watersheds for *mNSE*, and in eleven watersheds for *KGE* (Fig. 9), while with the forcing of observed rainfall negative indices were found in one, zero, and three watersheds, respectively (Fig. 5).

These results indicate that a small shift in time or space of modelled rainfall, in comparison to observed precipitation, can strongly modify the hydrologic response of small watersheds to extreme events. This is particularly evident in watersheds Pg and Mk, which are among the smallest and those characterized by the lowest average discharge in both events (Fig. 6, Fig. 7, Fig. S3, Fig. S4). Although their rainfall performance indices (Fig. 8) do not show particularly large errors (except a negative NSE for Mk in Nov 1994), streamflow fit indices present very negative values and streamflow PBIAS is very high as well (Fig. 9).The implementation of rainfall data correction or assimilation schemes could improve the forecasts of the atmospheric-hydrologic modelling chain, as demonstrated and discussed by previous studies (e.g., Avolio et al., 2019; Verri et al., 2017; Yucel et al., 2015). Recently, increasing efforts have been made to implement two-way coupled modelling systems, which were found to improve the overall skills of the modelling system (e.g., Senatore et al., 2015). However, the hydrologic component calibration is still usually performed based on observed precipitation data (e.g. Fersch et al., 2019; Givati et al., 2016).

The rainfall fields modelled by Zittis et al. (2017) and used in this study were downscaled from the ERA-Interim re-analysis dataset. The decision to use these modelled data was driven by the fact that ERA-Interim presents a resolution closer to that of existing forecasting, decadal prediction, and global climate models, therefore it resembles a modelling chain for forecasting applications and climate change projections (e.g., Reyers et al., 2019; Saha et al., 2014). For future studies ERA5, thanks to its finer resolution and the availability of ensemble members for uncertainty estimates, will be a valuable data source for improving the modelling chain over small (< 100 km$^2$) catchments.

**5.4 WRF-Hydro with observed and modeled precipitation evaluation at hourly scale**

Figure 10 shows the comparison between observed and modelled hourly hydrographs for three out of the seven watersheds that had modelled daily streamflow *NSE* larger than 0.5 in both calibration and validation events. The four watersheds that are not shown are Pg (hourly streamflow data not available), Pe, Ko, and Ni (rating curve not available for peak flow). Looking at the streamflow modelled with observed rainfall as forcing, hourly peaks are generally overestimated and the modelled streamflow response to rainfall appears more immediate (pulse-like) than the observed streamflow. The overestimation is more evident for the Nov-1994 validation event than for the Jan-1989 calibration event. In addition, the receding hydrograph is well modelled for the calibration event but not so well for the validation event. This result is similar to what was observed for daily streamflow and was attributed to the possible non-perfect reproduction of the model initial conditions and underestimation of

interflow. The fairly good post-peak simulations lead to reasonable hourly performance indices for the Jan-1989 event. However, even with an *NSE* of 0.80 and a *KGE* of 0.72 for watershed Ao, the 17.9 m³ s⁻¹ hourly peak flow was overestimated by 18%.

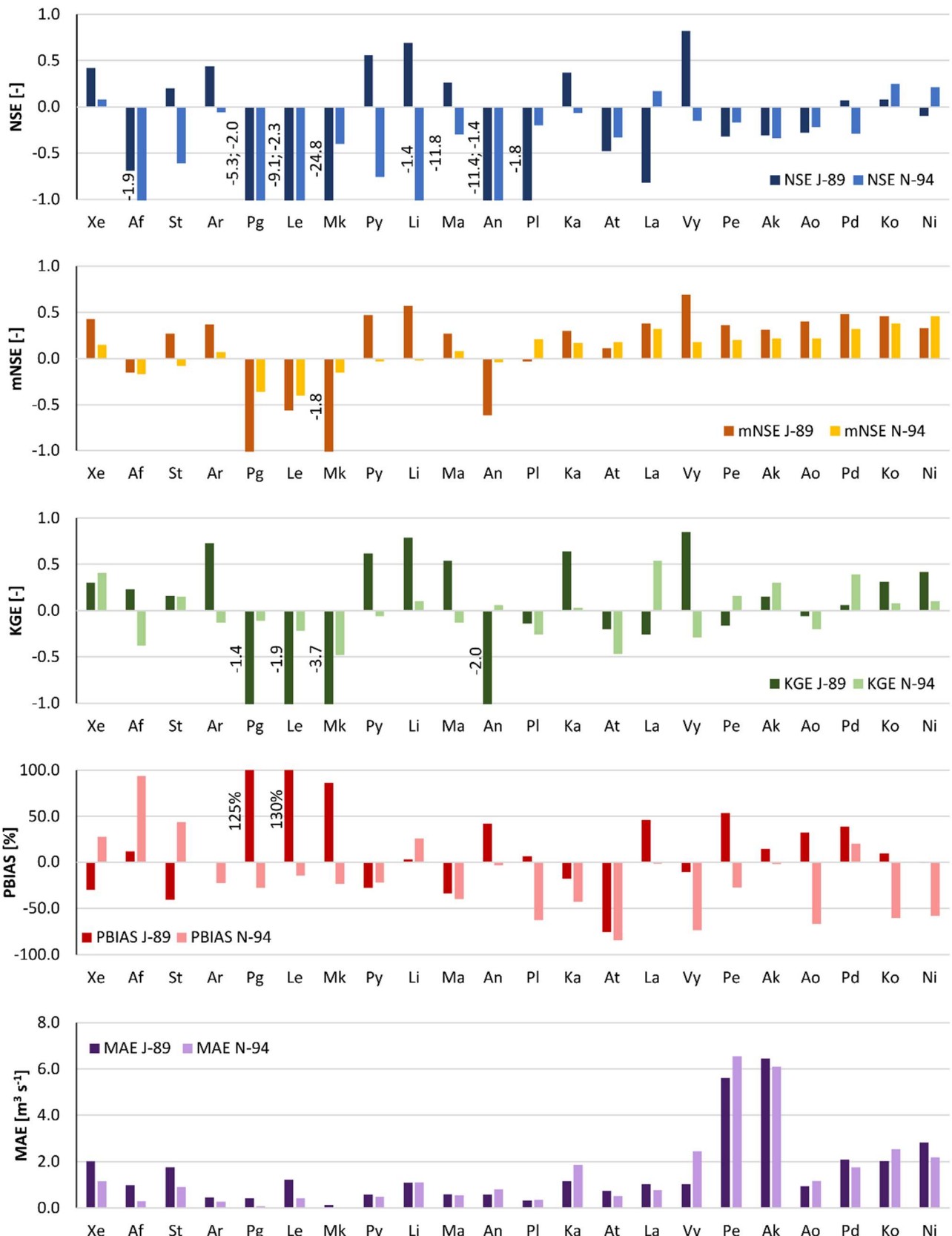

**Figure 9. Performance indices (NSE, Nash-Sutcliffe Efficiency; mNSE, modified Nash-Sutcliffe Efficiency; KGE, Kling-Gupta Efficiency; BIAS; MAE, Mean Absolute Error) calculated on daily streamflow resulting from WRF-modelled rainfall for the 22 watersheds using the calibrated set of parameters for both Jan 1989 (J-89) and Nov 1994 (N-94) events. For watershed short names refer to Table 1.**

The response of hourly streamflow to WRF-modelled rainfall shows similar behavior. The shape of the hydrographs is defined
by rainfall pulses, in terms of both time of response and size of peaks. Even more than for daily outputs, it is evident that small
differences in rainfall distribution and amount can cause large differences between observed and modelled streamflow (see
performance indices).

A possible improvement may be obtained by an increase in channel roughness coefficients. This would allow slower flow, and
a smoothing of the peaks. Especially in dry Mediterranean areas, characterized by streams with seasonal flow, the vegetation
(and consequently the roughness conditions) can be very different at the end of the dry period (vegetation grown within the
stream, dry understories and bushes and bare cropland overland) and in the middle of wet winter (water within the riverbed,
green vegetation cover overland). This could be described with the inclusion of a seasonal variation of channel and overland
roughness coefficients in the model. However, rainfall data with high spatial and temporal resolution would be essential to test
this model modification.

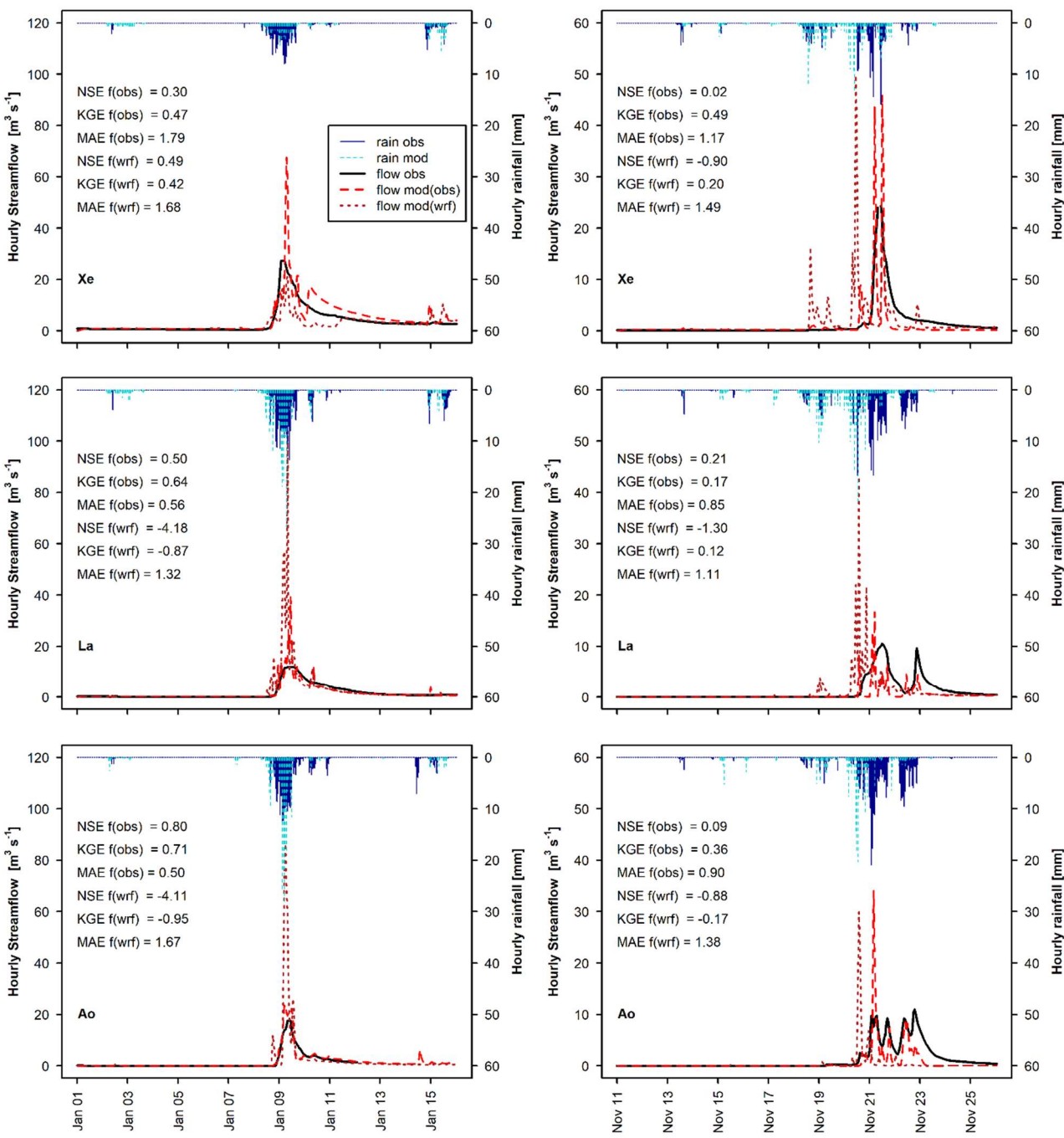

**Figure 10. Observed hourly hydrographs (flow obs) and hydrographs obtained with the calibrated WRF-Hydro model (flow mod)
forced with observed rainfall (rain obs) and with WRF-modelled rainfall (rain wrf) for both Jan 1989 (left) and Nov 1994 (right),
for three watersheds (see Table 1 for watershed short names); modelled flow performance indices (NSE, Nash-Sutcliffe Efficiency;
KGE, Kling-Gupta Efficiency; BIAS) are shown as well.**

# 6 Conclusion

This study evaluates streamflow simulations of the one-way coupled atmospheric-hydrologic model WRF-Hydro, forced with observed and WRF-modeled rainfall, during two extreme events, over 22 small mountain watersheds in Cyprus (area below 100 km$^2$). Following model calibration and validation with observed rain, the model was run with WRF-downscaled (1 × 1 km$^2$) re-analysis precipitation data (ERA-Interim). These forcing data represent best-performing hindcasts of two extreme rainfall events, i.e. a model product that is as similar as possible to reality and considered sub-optimal.

Overall, the selected four calibration parameters (REFKDT, Soil depth, the baseflow bucket exponent, and the maximum baseflow bucket capacity) were sufficient to obtain good model performance during model calibration in these steeply sloping and geologically complex watersheds. Sensitivity analysis showed that REFKDT can be calibrated beyond the suggested 0.5-5.0 range, having an effect on infiltration till a value of approximately 100.0. A Soil depth of 1.0 m, representative of the thin soils characterizing the study area, rather than the default value of 2.0 m, resulted in an average increase in *NSE* values of 0.14. Modifications of deep drainage coefficients and MODIS soil types based on geology reduced the peak flow overestimation by up to 40% in watersheds characterized by a fractured and very permeable bedrock. The overland roughness routing factor reduced the streamflow but showed a very limited effect on delaying flow. A straightforward calibration of the baseflow reservoir based on low flow fitting (exponent) and reservoir filling time (maximum capacity) was a good mean for obtaining a reasonable simulation of the hydrograph recession in most watersheds. Calculated daily *NSE* values were higher than 0.5 in 16 out of the 22 modeled watersheds in Jan 1989 (calibration) and in eight watersheds in Nov 1994 (validation). Negative *NSE* values were found in three watersheds located at high elevation where an underestimation of the snow fraction, computed by the LSM, may have occurred. Modelled snow height, and possible improvements deriving from the use of alternatives routines (e.g. Noah MP), should be checked with observed snow depth data, which were not available for this study.

The comparison of modelled and observed hourly streamflow showed that almost all peak flows were overestimated by the calibrated model. Modelled hourly streamflow fit the Jan 1989 hydrographs relatively well, but much less so the Nov 1994 discharges. This performance loss in Nov 1994 was due to a pulse-like behavior of the modeled streamflow related to an immediate response to rainfall, which could be attenuated by higher channel roughness coefficients.

Streamflow obtained with WRF-modelled rainfall forcing showed high discrepancies with observations, despite the good agreement between modelled and observed precipitation (average *NSE* of 0.83 and 0.49 for Jan 1989 and Nov 1994, respectively). This suggests that model calibration with modelled rainfall forcing is not optimal for small mountain watersheds and should be carefully evaluated if no other options are available. As a consequence, WRF rainfall forecasts may not be sufficiently accurate for predicting the location and size of specific floods of such small mountain watersheds. However, due to the relatively small errors in total precipitation (average relative difference over the 22 watersheds of 17% and for 20% Jan 1989 and Nov-1994 events, respectively) and simulated daily maxima (average relative difference over the 22 watersheds of 22% and 18% for Jan 1989 and Nov-1994 events, respectively), modelled rainfall data could be suitable for investigating the effect of climate change on extreme rainfall and flood events. From the results presented and discussed, it emerges that future studies could focus on various aspects of the modelling system to improve the simulation results of both precipitation and streamflow. Soil properties could be specifically calibrated for the study area. For a continuous, long-term streamflow analysis, an evaluation of the sensitivity of the baseflow reservoir parameters could be carried out. Also, the model could be improved by incorporating an option for time-dependent roughness coefficients to represent vegetation growth in ephemeral and intermittent streams in semi-arid environments. A model configuration with variable soil depths could also improve model performance, especially in mountain environments.

# 7 Acknowledgments

This research was funded by the BINGO project (Bringing INnovation to onGOing water management), European Union's Horizon 2020 Research and Innovation programme, under the Grant Agreement number 641739. For computation, this work was supported by the Cy-Tera Project (ΝΕΑ ΥΠΟΔΟΜΗ/ΣΤΡΑΤΗ/0308/31), which is co-funded by the European Regional Development Fund and the Republic of Cyprus through the Research Promotion Foundation. Authors would also like to thank the Water Development Department of Cyprus for data sharing and support.

## 540 Code/Data availability

WRF-Hydro is an open-source community model. The code and additional information can be found at https://ral.ucar.edu/projects/wrf_hydro/overview. WRF-Hydro simulated streamflow at the watershed outlets, for the two events (Jan 1989 and Nov 1994) and the two forcing (observed and modelled precipitation) are available on https://zenodo.org (DOI: 10.5281/zenodo.3952420).

## 545 Author contribution

Conceptualization, C.C., A.B., G.Z.; methodology, C.C., A.B; software, C.C., G.Z., I.S.; formal analysis, C.C.; investigation, C.C., I.S.; data curation, C.C., G.Z., I.S.; writing—original draft preparation, C.C.; writing—review and editing, A.B., G.Z, I.S., J.A.; supervision, A.B., J.A.; funding acquisition, A.B.

## Competing interests

The authors declare no competing interests.

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
