# Peer review of "Simulation of extreme rainfall and streamflow events in small Mediterranean watersheds with a one-way coupled atmospheric-hydrologic modelling system"

_Natural Hazards and Earth System Sciences, 2020_

## Referee Comment (RC1) · Anonymous Referee #1 · 30 Mar 2020

The paper of Camera et al. presents a complete hydrometeorological reanalysis of two high impact events in Cyprus island (Eastern Mediterranean) addressing the challenge of effective reconstruction of such kind of events for small to very small catchments (ranging from 5 to less than 100 km2 in this study). Overall, the paper presents a detailed and complete exercise, which adds another piece to the puzzle, benefiting from the availability of increasingly advanced modelling systems at all scales of analysis. Furthermore, the analysis is performed over an extraordinarily important area for Cyprus water resources, using a considerable set of discharge data and also dealing

(even though partially) with the challenging issue of hydrological modelling in a mountain environment with rock fractures.

I suggest three main improvements to the paper, listed below, and have some other minor comments. I hope my comments are helpful to further enhance the quality of the paper.

- My first main comment concerns the GCM data source (i.e., ERA-Interim). I acknowledge that this study inherits the work done by Zittis et al. (2017), but this global reanalysis is now replaced by the ERA5 reanalysis. This point is important, also given the fact that ERA5 offers ensemble members, which could be very usefully used exactly for the problem analysed (i.e., hydrometeorological chains targeted to small and very small catchments). I ask the authors to deal with this point, of course not requiring new simulations with ERA5, but discussing it.

- Furthermore, I have some concerns about the calibration and use of the bucket model. In general, my idea is that the baseflow bucket model could not be so important for such short-time events. Indeed, the case studies analysed are rather impulsive. Furthermore, I think that the effects of the bucket model are somehow misinterpreted (please refer to a specific comment below). I suggest the authors revise and comment on their choice of calibrating in detail the baseflow bucket model.

- Finally, I believe the authors can go more into details analysing the catchments with rock fractures, which show too low performances that should be increased somehow (please refer to specific comments below).

Minor/specific comments

Abstract: stating that "few studies evaluate the hydrologic performance etc. . . ." is a bit debatable concept (e.g., few with respect to what?). This statement is different from a similar one on L81, where the authors specify that they are referring to WRF-Hydro. I would start the manuscript with a stronger sentence. Furthermore, in the Abstract the

fact that 1989 events are used for calibration and 1994 events for validation should be stated more clearly.

L46 (and throughout the text): I would write "As summarized by Rummler et al. (2019)" rather than "As summarized by (Rummler et al., 2019)".

L85: it looks like the events are much shorter. Including the spin-up period in this time interval could be misleading.

Fig. 1: I suggest the authors focus more on the WRF-Hydro domain, which could be represented with a larger scale (so that also other information, e.g., location of rain gauge stations and reservoirs, can be added). Location of the WRF-Hydro domain in Cyprus island could be shown with another small map in the figure.

Table 1: A clear geological description is ok, but I would also highlight some essential geographical/morphological features, such as area, channel length, etc. Maybe authors can move some piece of information from Table 4 or just repeat it.

L121: the problem of getting a reliable rating curve is rather common. More details about the "appropriate" rating curves used would be useful.

Eq. 6: the variable Z should be explicitly defined

L218: information about average soil moisture would make more sense if information about soil type was provided

L228: 1500 cells should be 1500 x 100 x 100 = 15M m2, that is 15 km2 (it should be better stated explicitly). However, in Table 4 there are some catchments with area lower than this threshold.

L267: at a time

Fig. 4 and elsewhere: to compare the performances of the model system for the two events, probably percent bias and MAE are more appropriate indices

LL320-333: [this comment refers to the main comment about dealing with rock fractures] from this paragraph, it's not clear if the problem is mainly related to the snow model in the LSM or the not good representation of the geological features. I would favour the second hypothesis, and I think that some test should be performed (and shown) by the authors increasing drainage.

LL335-339 and Figs. 5-6: the Y scale for watershed Mk is not appropriate (much higher maximum value than needed). The comment about watershed ST does not correspond to what I can see in the Figures.

L343: [this comment refers to the main comment about the groundwater bucket model] For Ak, it's not a problem of baseflow, but of recession, which is typically a problem concerning especially interflow (i.e., quicker contribution than baseflow).

L344: the peak looks not so well simulated in Ak

L68: passing -> moving?

LL372-374: these sentences are confusing, especially if compared with LL351-353, which seem to refer to the same comparison. Not clear what the authors mean when they state that bias "on average increased by 8.6 times"

L395: the three watersheds

L400: decent -> reasonable? Besides, again I don't think it's a matter of baseflow

L412: probably, increasing overland roughness coefficient could be also a way for improving interflow and, therefore, the simulation of the falling limb of the hydrograph

LL442-443: please contextualize better this sentence

---

## Referee Comment (RC2) · Anonymous Referee #2 · 28 Apr 2020

This paper presents a modeling work on 22 small watersheds using WRF-hydro. the model is forced with modeled WRF data and observe precipitation for 2 periods in January 1989 and November 1994. The authors concluded that using WRF precipitation may not be suitable for hydrological studies in small mountain basins although they are still useful for long term studies.

General comment:

I am not quite sure what is the research question that the authors are trying to answer.

[Figure]

If it is WRF-hydro ability to simulate streamflow, I think that is widely covered in the literature review, but it may be important a benchmarking in this specific area. If it is the advantage of using observed precipitation, I think that it is not necessary to write a paper about it since it is well known that if there are observations available, it is better to use them over modeled data or to correct the modeled data. Therefore, it is not clear to me what is the actual contribution that the authors are trying to deliver.

The problem with modeled precipitation is because the WRF model does not work? Or the modeler did not implement it correctly? When the model and observe precipitation does not compare well, why did you use it anyway? I think that using incorrect input will certainly result in poor performance. But jumping from there to conclude that we should not use modeled precipitation is a big stretch. I think that the paper should focus on the performance of WRF-hydro. The model performance is not affected if the precipitation is observed or modeled since you are using it uncoupled.

Specific Comment

• First line abstract: "Few studies evaluate the hydrologic performance of coupled atmospheric-hydrologic models when forced with observed rainfall and even fewer when forced with modeled precipitation." This is not quite true, there is extensive literature on this topic. If you are specifically referring to WRF-hydro you should state that.

• The first paragraph of the introduction is related to the land-atmosphere feedbacks, which is not the case of this study, so I suggest eliminating it or to refocus it to the topic of the paper

• What about the bucket model parameter sensitivity? There is no indication of the interaction of those in the uncertainty analysis.

• Is there any evaluation of precipitation disaggregation eq 1 and 2?

• In Figure 4, there a series of inconsistencies between the performance of the

model between the calibration and evaluation period. Is there any explanation for that? In particular basins, MA, An, Pi, Ka, where you have really bad results during the calibration but still validate the models and got very good results.

• Also, it would be better to use percentage bias instead of bias alone to have a more general indicator of biases.

• Figures 5 and 6, it is very hard to see the differences in precipitation. And then, why to use incorrect model precipitation at all.

• If we look at figures 7 and 8, we can see that WRF-hydro does a really bad job (Figure 8) in basins where WRF precipitation is has good performance (Figure 7), can you explain why?

• In the conclusion you mention this "Streamflow obtained with WRF-modelled rainfall forcing showed high discrepancies with observations, despite the good agreement between modeled and observed precipitation (average NSE of 0.83 and 0.49 for Jan 1989 and Nov 1994, respectively)." Did you try to calibrate the model with the modeled precipitation data and evaluate the observation?

---

## Author Comment (AC1) · 19 Jun 2020

The paper of Camera et al. presents a complete hydrometeorological reanalysis of two high impact events in Cyprus island (Eastern Mediterranean) addressing the challenge of effective reconstruction of such kind of events for small to very small catchments (ranging from 5 to less than 100 km$^2$ in this study). Overall, the paper presents a detailed and complete exercise, which adds another piece to the puzzle, benefiting from the availability of increasingly advanced modelling systems at all scales of analysis. Furthermore, the analysis is performed over an extraordinarily important area for Cyprus water resources, using a considerable set of discharge data and also (even though partially) with the challenging issue of hydrological modelling in a mountain environment with rock fractures. I suggest three main improvements to the paper, listed below, and have some other minor comments. I hope my comments are helpful to further enhance the quality of the paper.

1. My first main comment concerns the GCM data source (i.e., ERA-Interim). I acknowledge that this study inherits the work done by Zittis et al. (2017), but this global reanalysis is now replaced by the ERA5 reanalysis. This point is important, also given the fact that ERA5 offers ensemble members, which could be very usefully used exactly for the problem analysed (i.e., hydrometeorological chains targeted to small and very small catchments). I ask the authors to deal with this point, of course not requiring new simulations with ERA5, but discussing it.

   The decision to use ERA-Interim was driven by the previous work of Zittis et al. (2017) and also by the fact that we wanted to downscale a re-analysis dataset that was closer to the resolution of existing forecasting, decadal prediction, and global climate models in order to resemble a realistic modelling chain for forecasting applications. Moreover, ERA5 is not yet in a very mature stage, as evidenced from the emails alerting users from time to time to the presence of errors in the database. Also, in some cases re-runs are released for some years because of simulation errors (Simmons et al., 2020). However, we agree that ERA5 represents an opportunity for future improvement of the model skills. We have added few lines in the abstract and a discussion of the matter in Section 5.3.

   **Abstract, Line 18-19:** "This set up resembles a realistic modelling chain for forecasting applications and climate projections".

   **Results, section 5.3 WRF-Hydro simulations with modeled precipitation, Line 481-486:** "The rainfall fields modelled by Zittis et al. (2017) and used in this study were downscaled from the ERA-Interim re-analysis dataset. The decision to use these modelled data was driven by the fact that ERA-Interim presents a resolution closer to that of existing forecasting, decadal prediction, and global climate models, therefore it resembles a modelling chain for forecasting applications and climate change projections (e.g., Reyers et al., 2019; Saha et al., 2014). For future studies ERA5, thanks to its finer resolution and the availability of ensemble members for uncertainty estimates, will be a valuable data source for improving the modelling chain over small (< 100 km$^2$) catchments".

2. Furthermore, I have some concerns about the calibration and use of the bucket model. In general, my idea is that the baseflow bucket model could not be so important for such short-time events. Indeed, the case studies analysed are rather impulsive. Furthermore, I think that the effects of the bucket model are somehow misinterpreted (please refer to a specific comment below). I suggest the authors revise and comment on their choice of calibrating in detail the baseflow bucket model.

The hydrograph recession is made up of delayed surface runoff, interflow (lateral subsurface flow from the soil) and baseflow (groundwater). As suggested by Reviewer 1 in specific comment 20, we investigated the possibility to fit it by calibrating the overland roughness routing factor (OVRGH). We tested the sensitivity of OVRGH and we noticed that the parameter wasn't helpful in redistributing discharge, it was just increasing or decreasing it without modifying the shape of the hydrograph (new Fig. 3 and Fig. 4). In addition, for our runs we had already set OVRGH=1, which according to many authors is the maximum possible value that can be assigned to the parameter (Yucel et al., 2015; Verri et al., 2017). Therefore, we tried to capture the hydrograph recession better by increasing baseflow through the calibration of the reservoir (bucket) maximum volume ($Z_{max}$) and exponent ($\alpha$). For $Z_{max}$, we aimed to set its value so that the reservoir could be filled between 10 January at h. 00:00 and 11 January at h. 12:00, indicatively within few hours and 2 days after the peak rainfall. The model redistributes the deep percolation exceeding the reservoir volume between the channel cells of the corresponding watersheds. For those watersheds that highly overestimated the baseflow due to spilling out of the groundwater reservoir, we further increased $Z_{max}$. For the exponent, we calibrated it fitting the pre-peak hydrograph. Details regarding how we modified the manuscript to incorporate these analyses and their results are given in the answers to the specific comments.

3. Finally, I believe the authors can go more into details analysing the catchments with rock fractures, which show too low performances that should be increased somehow (please refer to specific comments below).

We tackled the problem from two sides. First, we modified the terrain slope categories (SLOPECAT) map and consequently the SLOPE coefficients (controlling deep drainage) based on geology. For gabbro and ultramafic rock types we forced a SLOPECAT resulting in a SLOPE coefficient equal to 1 (i.e., the maximum possible value) and therefore in a maximization of the drainage from the soil column to the groundwater reservoir. Second, based on geology and field observations (Camera et al., 2018), we modified the soil type map as well. The MODIS database, which was used for soil characterization, attributes a uniform clay loam soil texture to the Troodos Mountains. However, we have observed that at the higher elevations, where predominant geology is gabbro and ultramafic rocks, soils show a gravelly sandy loam texture (Camera et al., 2018). Therefore, we modified the MODIS map, attributing a sandy loam soil type for cells characterized by gabbro and ultramafic rocks. In the WRF-Hydro model, soil properties are linked to soil type. For the cell involved, this change of soil type resulted in a modification (among other properties) of the saturated hydraulic conductivity from 2.45E-6 m/s to 5.32E-6

m/s. Before applying these changes, we investigated the sensitivity of the saturated hydraulic conductivity (Ks), which was found to be a sensitive parameter (see new Fig. 3).

Despite our efforts to maximize infiltration and deep drainage to reduce the hydrograph peak, the model still overestimated the observed flow in the high elevation watersheds. Looking at observed temperature time series, it is likely that part of the precipitation on the mountains occurred as snow during the January 1989 event. However, we do not have observed snow height data. The WRF atmospheric forcing data, which was used coupled with the observed precipitation, slightly underestimates the temperature on the top of the mountains (i.e., the model is colder than reality). Thus, it does not seem to be a modelled temperature issue. The land surface model converts precipitation into snow and snow into melt water through a radiation- and temperature-based routine. The simulated snow depth and snow-water equivalent during the event of January 1989 might be lower than expected. Another indication sustaining this hypothesis is that for the event of November 1994 the model slightly underestimates the hydrograph peak. Details regarding how we modified the manuscript to incorporate these analyses are given in the answers to the specific comments.

**Minor/specific comments**

1. Abstract: stating that "few studies evaluate the hydrologic performance etc. . . . " is a bit debatable concept (e.g., few with respect to what?). This statement is different from a similar one on L81, where the authors specify that they are referring to WRF-Hydro. I would start the manuscript with a stronger sentence. Furthermore, in the Abstract the fact that 1989 events are used for calibration and 1994 events for validation should be stated more clearly.
   We have modified the first sentence of the abstract and have added the reference to calibration and validation for the two events of January 1989 and November 1994 as follows.

   **Abstract, Line 12-13:** "Coupled atmospheric-hydrologic systems are increasingly used as instruments for flood forecasting and water management purposes, making the performance of the hydrologic routines a key indicator of the model functionality".

   **Abstract, Line 19-20:** "Streamflow was modelled during extreme rainfall events that occurred in January 1989 (calibration) and November 1994 (validation) over 22 mountain watersheds".

2. L46 (and throughout the text): I would write "As summarized by Rummler et al. (2019)" rather than "As summarized by (Rummler et al., 2019)".
   Thanks for spotting it, we modified as suggested and we searched for similar occurrences throughout the manuscript.

3. L85: it looks like the events are much shorter. Including the spin-up period in this time interval could be misleading.
   To clarify this point, we modified the manuscript as follows.

   **Introduction, Line 89-95:** "The focus is on two extreme events that occurred over 22 small watersheds, located in the Troodos Mountains of Cyprus, between 8-10 January 1989 and 20-22

November 1994. The main objectives are: (i) to calibrate the uncoupled WRF-Hydro model for simulating extreme events in Cyprus with observed precipitation; and (ii) to evaluate the model performance when forced with WRF-downscaled (1 × 1 km$^2$) re-analysis precipitation data (ERA-Interim). The model runs covered two 15-day periods (1-16 January and 11-26 November) to include a short spin-up of the WRF-Hydro routines and the simulation and evaluation of the receding limb of the hydrograph".

4. Fig. 1: I suggest the authors focus more on the WRF-Hydro domain, which could be represented with a larger scale (so that also other information, e.g., location of raingauge stations and reservoirs, can be added). Location of the WRF-Hydro domain in Cyprus island could be shown with another small map in the figure.
We have modified Fig. 1 according to the suggestions.

5. Table 1: A clear geological description is ok, but I would also highlight some essential geographical/morphological features, such as area, channel length, etc. Maybe authors can move some piece of information from Table 4 or just repeat it.
We added area and channel length in Table 1 and left all the other variables in Table 4 as they were in the previous version of the manuscript.

6. L121: the problem of getting a reliable rating curve is rather common. More details about the "appropriate" rating curves used would be useful.
We have added the following.

**Data, section 3.1 streamflow data, Line 124-130:** "For the 22 watersheds, daily discharge data (m$^3$ s$^{-1}$) from streamflow stations of the Cyprus Water Development Department for the period 1980-2010 were analyzed. In addition, the original continuous hydrograph charts (water levels) of 16 of the 22 streamflow stations from the Water Development Department, for the Jan-1989 and Nov-1994 events, were scanned and manually digitized through the GetData Graph Digitizer software (http://getdata-graph-digitizer.com). The digitized water levels were interpolated to obtain values precisely every 15 minutes (00.00, 00.15, 00.30, 00.45, 01.00….) and converted to discharge with the appropriate rating curve of the station. The streamflow stations and rating curves are maintained by the Water Development Department through frequent observations".

7. Eq. 6: the variable Z should be explicitly defined
We have added an equation (eq. 7) to define Z. The manuscript has been modified as follows.

**Modelling setup, section 4.1 WRF-Hydro model description, Line 208-218:** "The second solution consists of calculating a baseflow discharge [m$^3$ s$^{-1}$] ($Q_{bf}$) by means of an exponential bucket model, described by the following equation:

$$Q_{bf} = C \cdot \left( e^{a \cdot \frac{Z}{Z_{max}}} - 1 \right), \tag{6}$$

where $C$ is the bucket coefficient [$m^3$ $s^{-1}$], $a$ is the bucket model exponent [-], $Z_{max}$ is the maximum bucket level [m], and $Z$ [m] is the bucket level at a certain time step. The user defines the $C$, $a$ and $Z_{max}$ parameters for each sub-watershed, together with a $Z_{ini}$ [m] parameter to initialize the water storage in the bucket groundwater reservoir. At each time step the $Z$ value is updated first adding the deep drainage contribution ($Perc$) and subsequently subtracting $Q_{bf}$:

$$Z_t = Z_{t-1} + \sum_{n=1}^{n=ncells} Perc_n - \frac{Q_{bf} \cdot DT \cdot 3600}{A} \tag{7}$$

where A is the area of the sub-watershed [$m^2$], DT the model time step [day], n is the index for the sub-watershed cells, and ncells represents the number of cells of the sub-watershed. Similar to the first solution, $Q_{bf}$ is equally redistributed to channel segments. If $Z$ equals or exceeds $Z_{max}$, all deep drainage is transferred to the channel network".

8. L218: information about average soil moisture would make more sense if information about soil type was provided
We have modified the manuscript specifying the soil type as measured during experiments. Also, we added how we modified the original MODIS soil map to take into consideration the high permeable soils of the upper mountains (see also answer to general comment 3):

**Methods, section 4.2 WRF-Hydro Parameterization, Line 234-242:** "Experimental data (Camera et al., 2018) show that in these conditions soil moisture for a gravelly sandy loam at 1300 m a.s.l. in the Troodos Mountains can vary between 0.10 and 0.15 $m^3$ $m^{-3}$. Therefore, the WRF-derived initial soil moisture values for November were halved.
Land use and vegetation cover data were derived from the MODIS dataset through the WRF Pre-Processing System. According to the MODIS dataset, the Troodos Mountains has a uniform clay loam texture. However, field observations at higher elevation in the mountains, where the predominant lithologies consist of gabbro and ultramafic rocks, showed a gravelly sandy loam texture (Djuma et al., 2020; Camera et al., 2018; Cyprus Geological Survey Department, 1995). In addition, it is known that the Troodos gabbro is very weathered and therefore permeable (Christofi et al., 2020). Therefore, a sandy loam soil type was assigned to these areas.".

9. L228: 1500 cells should be 1500 x 100 x 100 = 15M $m^2$, that is 15 $km^2$ (it should be better stated explicitly). However, in Table 4 there are some catchments with area lower than this threshold.
Right, that data was wrongly reported. The threshold is 250 cells (2.5 $km^2$). It is now clearly stated in the manuscript.

**Methods, section 4.2 WRF-Hydro Parameterization, Line 245-246:** "For the channel grid, a flow accumulation threshold of 1500 250 cells (2.5 $km^2$) was adopted".

10. L267: at a time
Modified as:

**Methods, section 4.2 WRF-Hydro Parameterization, Line 265-267:** "The initial level of the conceptual reservoir ($Z_{ini}$) was set as a fraction of the maximum level ($Z_{max}$), based on the saturation degree of the deepest soil layer at the end of the 15-day WRF spin-up period".

11. Fig. 4 and elsewhere: to compare the performances of the model system for the two events, probably percent bias and MAE are more appropriate indices
We have modified Fig. 4, Fig. 7, and Fig. 8 substituting BIAS with percent bias (PBIAS). Figure numbering changed because we added a new Fig. 4 for the sensitivity analysis results, so they are now Fig. 5, Fig. 8, and Fig. 9.

12. LL320-333: [this comment refers to the main comment about dealing with rock fractures] from this paragraph, it's not clear if the problem is mainly related to the snow model in the LSM or the not good representation of the geological features. I would favour the second hypothesis, and I think that some test should be performed (and shown) by the authors increasing drainage.
As explained in the answer to the general comment 3, we have modified the parameter controlling deep drainage and the soil type based on geology (increased deep drainage and coarser soil for areas with gabbro and ultramafic rocks). We have incorporated in the sensitivity analysis one run with the modified deep drainage and three runs with different saturated hydraulic conductivity values, relative to different soil textures in the soil parameter tables. We have noticed a high sensitivity of saturated hydraulic conductivity and a rather low sensitivity of the deep drainage parameter. In the final model parameterization, we considered the results of the sensitivity analysis. In detail, we modified the manuscript as follows.

[revised manuscript text omitted]

13. LL335-339 and Figs. 5-6: the Y scale for watershed Mk is not appropriate (much higher maximum value than needed). The comment about watershed St does not correspond to what I can see in the Figures.
    We modified the Y-scale of Mk in all figures and the comments related to both watersheds as follows.

    **Results, section 5.2 WRF-Hydro calibration and validation, Line 404-409:** "Mk is the only watershed showing higher rainfall and flow peaks towards the end of the Jan-1989 event rather than in the middle. The model slightly underestimates the flow peak occurred on January 9th and overestimates the flow at the end of the simulation period. For St, the model reacts sharply to precipitation input, simulating well the flow peak occurred on January 9th but overestimating the flow at end of the simulation period of the Jan-1989 event and above all the peak of the Nov-1994 event, therefore affecting the performance scores".

14. L343: [this comment refers to the main comment about the groundwater bucket model] For Ak, it's not a problem of baseflow, but of recession, which is typically a problem concerning especially interflow (i.e., quicker contribution than baseflow).
    Our bedrock is very fractured without a continuous groundwater table and we have predominantly shallow soils. It is difficult to distinguish between interflow and baseflow. We have observed slow dripping from the bedrock into upstream channels after large rainfall events. We also have streams that discharge to the bedrock with streamflow again recurring further downstream. Thus, we do have a streamflow recession made up of a combination of processes. As noted in general comment nr. 2, the OVRGH parameter influences the total discharged volume but not the shape of the hydrograph. Therefore, to better fit the post-peak shape of the hydrograph, we focused on baseflow calibration. To monitor the baseflow effect, we added four figures as supplementary material (Fig. S1 – S4), in which we showed the hydrographs for all watersheds together with the baseflow contribution, for both events and both observed and modelled rainfall as forcing. Fig. S1 and Fig. S3 show hydrographs for Jan-1989 event forced with observed and modelled rainfall, respectively. Fig. S2 and Fig. S4 show hydrographs for Nov-1994 event forced with observed and modelled rainfall, respectively. To incorporate these analyses, we modified the manuscript as follows.

[revised manuscript text omitted]

15. L344: the peak looks not so well simulated in Ak
   We agree. We have modified the manuscript as follows.

   **Results, section 5.2 WRF-Hydro calibration and validation, Line 410-418:** "In the eastern part of the modelling domain (La to Ni), for the calibration event both initial baseflow and the discharge peak are well modelled in all watersheds (Fig. 6). Differences between observed and simulated hydrographs can be observed in the post-peak, for watersheds Ak, Pe (Fig. S1), Ko and Ni. Ak and Pe present a very high peak flow ($> 50$ m$^3$ s$^{-1}$) and an underestimation of the receding limb of the hydrograph in the following days, which causes the negative *PBIAS* and high *MAE* values visible in Fig 5. In the case of Ko and Ni, the receding limb shows a little overestimation. For the validation event (Fig. 7), the peak is well simulated in Pe and Ao, slightly overestimated in Ak and Pd, underestimated in La, Vy, Ko, and Ni (Pe and Pd, Fig. S2). In the post peak phase, the simulated hydrographs show negative biases in comparison to the observed ones in all watersheds".

16. L368: passing -> moving?
   Modified as suggested.

17. LL372-374: these sentences are confusing, especially if compared with LL351-353, which seem to refer to the same comparison. Not clear what the authors mean when they state that bias "on average increased by 8.6 times"
   The first lines described rainfall, while the second group described streamflow. Throughout section 5.3 we have now modified the text so that it is explicitly said if the performance indices

refer to precipitation or streamflow. We introduced PBIAS as a replacement of BIAS so we modified the unclear sentence.

**Line 461-463:** "The absolute value of flow *PBIAS* decreased in seven watersheds (Af, Li, Pl, Vy, Ak, Ko, Ni) but on average increased by 21.5% (96.6% in Pg and 120.3% in Le)".

18. L395: the three watersheds

Thanks for spotting it. Changing some parameter during calibration the watersheds became four (**Line 495** in the manuscript with track changes).

19. L400: decent -> reasonable? Besides, again I don't think it's a matter of baseflow

We modified the text discussing the receding limb of the hydrograph in general and not baseflow only.

**Results, section 5.4 WRF-Hydro with observed and modeled precipitation evaluation at hourly scale, Line 500-503:** "In addition, the receding hydrograph is well modelled for the calibration event but not so well for the validation event. This result is similar to what was observed for daily streamflow and was attributed to the possible non-perfect reproduction of the model initial conditions and underestimation of interflow. The fairly good post-peak simulations lead to reasonable hourly performance indices for the Jan-1989 event.".

20. L412: probably, increasing overland roughness coefficient could be also a way for improving interflow and, therefore, the simulation of the falling limb of the hydrograph

Please refer to answer to previous comments regarding overland roughness and interflow (general comment 2, minor comments 14, 15, 19).

21. LL442-443: please contextualize better this sentence

We have modified the sentence.

**Conclusion, Line 551-557:** "This suggests that model calibration with modelled rainfall forcing is not optimal for small mountain watersheds and should be carefully evaluated if no other options are available. As a consequence, WRF rainfall forecasts may not be sufficiently accurate for predicting the location and size of specific floods of such small mountain watersheds. However, due to the relatively small errors in total precipitation (average relative difference over the 22 watersheds of 17% and for 20% Jan 1989 and Nov-1994 events, respectively) and simulated daily maxima (average relative difference over the 22 watersheds of 22% and 18% for Jan 1989 and Nov-1994 events, respectively), modelled rainfall data could be suitable for investigating the effect of climate change on extreme rainfall and flood events".

---

## Author Comment (AC2) · 19 Jun 2020

This paper presents a modeling work on 22 small watersheds using WRF-hydro. The model is forced with modeled WRF data and observe precipitation for 2 periods in January 1989 and November 1994. The authors concluded that using WRF precipitation may not be suitable for hydrological studies in small mountain basins although they are still useful for long term studies.

**General comment**

1. I am not quite sure what is the research question that the authors are trying to answer. If it is WRF-hydro ability to simulate streamflow, I think that is widely covered in the literature review, but it may be important a benchmarking in this specific area. If it is the advantage of using observed precipitation, I think that it is not necessary to write a paper about it since it is well known that if there are observations available, it is better to use them over modeled data or to correct the modeled data. Therefore, it is not clear to me what is the actual contribution that the authors are trying to deliver. The problem with modeled precipitation is because the WRF model does not work? Or the modeler did not implement it correctly? When the model and observe precipitation does not compare well, why did you use it anyway? I think that using incorrect input will certainly result in poor performance. But jumping from there to conclude that we should not use modeled precipitation is a big stretch. I think that the paper should focus on the performance of WRF-hydro. The model performance is not affected if the precipitation is observed or modeled since you are using it uncoupled.

We used the best possible approach to model convective rainfall events and the results obtained show a good agreement with the observed fields. They compare well. Therefore, we consider our modelled rainfall input as a sub-optimal input. Although the high quality of it, still the small errors in the rainfall, in such small watersheds, propagate in the streamflow deeply affecting the performance. We have modified the introduction and the conclusion to state in a clear way that the value of the results is limited to small watershed (below 100 km²) and that model calibration carried out with modelled data is not optimal, not that it shouldn't be performed at all. Also, we suggest that WRF rainfall forecasts may not be sufficiently accurate for predicting the location and size of the floods of such watersheds thinking of implementing a similar system as an operational flood forecasting tool.

**Introduction, Line 87-89:** "Model performance loss due to differences between observed and modelled rainfall is rarely discussed. Also, little attention has been given to small watersheds (area below 100 km²), which are often ungauged and prone to flash floods. This study aims to address this gap".

**Conclusion, Line 523-527:** "This study evaluates streamflow simulations of the one-way coupled atmospheric-hydrologic model WRF-Hydro, forced with observed and WRF-modeled rainfall, during two extreme events, over 22 small mountain watersheds in Cyprus (area below 100 km²). Following model calibration and validation with observed rain, the model was run with WRF-downscaled (1 × 1 km²) re-analysis precipitation data (ERA-Interim). These forcing data represent best-performing hindcasts of two extreme rainfall events, i.e. a model product that is as similar as possible to reality and considered sub-optimal".

**Conclusion, Line 551-557:** "This suggests that model calibration with modelled rainfall forcing is not optimal for small mountain watersheds and should be carefully evaluated if no other options are available. As a consequence, WRF rainfall forecasts may not be sufficiently accurate for predicting the location and size of specific floods of such small mountain watersheds. However, due to the relatively small errors in total precipitation (average relative difference over the 22 watersheds of 17% and for 20% Jan 1989 and Nov-1994 events, respectively) and simulated daily maxima (average relative difference over the 22 watersheds of 22% and 18% for Jan 1989 and Nov-1994 events, respectively), modelled rainfall data could be suitable for investigating the effect of climate change on extreme rainfall and flood events".

**Specific Comments**

1. First line abstract: "Few studies evaluate the hydrologic performance of coupled atmospheric-hydrologic models when forced with observed rainfall and even fewer when forced with modeled precipitation." This is not quite true, there is extensive literature on this topic. If you are specifically referring to WRF-hydro you should state that.
   We have modified the first sentence of the abstract as follows.

   **Abstract, Line 12-13:** "Coupled atmospheric-hydrologic systems are increasingly used as instruments for flood forecasting and water management purposes, making the performance of the hydrologic routines a key indicator of the model functionality".

2. The first paragraph of the introduction is related to the land-atmosphere feedbacks, which is not the case of this study, so I suggest eliminating it or to refocus it to the topic of the paper
   We have refocused the paragraph to the topic of the paper by adding a sentence at the end of it.

   **Introduction, Line 39-40:** "However, recently authors have started to see these systems as instruments for flood forecasting, making the performance of the hydrologic routines a key indicator of the model quality (Givati et al., 2016; Maidment 2017)".

3. What about the bucket model parameter sensitivity? There is no indication of the interaction of those in the uncertainty analysis.
   Yes, good point. We conducted the sensitivity analysis on the parameters mainly influencing the rainfall runoff-infiltration partitioning (including few sensitivity runs regarding the overland roughness coefficient that we added during the review), with the baseflow bucket model switched off. For the calibration, we found that we could improve the simulations with a straightforward tuning of the baseflow parameters based on its filling time and the fit of the pre-peak hydrograph. We agree that a sensitivity analysis of the baseflow parameters is useful, but it would be more suitable to do this for a continuous, long-term streamflow analysis. We have added a recommendation about this in the conclusions.

   **Conclusion, Line 560-561**: "For a continuous, long-term streamflow analysis, an evaluation of the sensitivity of the baseflow reservoir parameters could be carried out".

4. Is there any evaluation of precipitation disaggregation eq 1 and 2?

We derived hourly fields with the presented disaggregation method and with simple IDW interpolation, since the method worked best for daily local events (Camera et al., 2014). We forced WRF-Hydro with both datasets and obtained a better fit of streamflow with the disaggregated one. In addition, looking at the 5-day cumulated rainfall fields calculated from the two hourly datasets, for the days around the peak precipitation for the two events of interest, we noticed a more plausible areal distribution for the disaggregation method (Fig. 1R). In addition, this method allows to preserve the mass balance between the daily and the hourly dataset. We did not include this explanation in the manuscript for conciseness.

[Figure]

**Fig. 1R: comparison of 5-day cumulated rainfall around the precipitation peak of the two events of interest (Jan 1989 and Nov 1994) obtained with two different interpolation methods, Inverse Distance Weighting (IDW) and disaggregation of daily to hourly values.**

5. In Figure 4, there a series of inconsistencies between the performance of the model between the calibration and evaluation period. Is there any explanation for that? In particular basins, Ma, An, Pi, Ka, where you have really bad results during the calibration but still validate the models and got very good results.

This relates to some of the comments Reviewer #1 made, too. For us, it is partly related to the geological characteristics of those watersheds and partly to the fact that they are located at high elevation and part of the precipitation during Jan-89 event occurred as snow. We managed to slightly improve the hydrograph simulation of these watershed (still they are not optimal)

14

modifying deep drainage and soil properties based on geology. We have modified the SLOPECAT map and consequently the SLOPE coefficients (controlling deep drainage) based on geology. For gabbro and ultramafic rock types we forced a SLOPECAT resulting in a SLOPE coefficient equal to 1 (i.e., the maximum possible value) and therefore a maximization of the drainage from the soil column to the groundwater bucket. Also, we modified the soil type of the area from clay loam (MODIS database) to sandy loam, based on field evidence. The overestimation of the peak got reduced up to 40% but still an overestimation remained. The same model parameterization results in positive NSE and negative BIAS for the Nov-1994 event. This combination of results led us to believe that the main issue is an underestimation of the snow. Please refer to the answers to the general comment 3 and specific comment 12 of Reviewer #1 for details on the added analyses and manuscript modifications.

6. Also, it would be better to use percentage bias instead of bias alone to have a more general indicator of biases.
   We modified the manuscript accordingly.

7. Figures 5 and 6, it is very hard to see the differences in precipitation. And then, why to use incorrect model precipitation at all.
   We changed the color of modelled rainfall, we hope figures are clearer than in the previous version. Figures numbering changed because we added an extra figure to discuss the sensitivity analysis results. The mentioned figures are now Fig. 6 and Fig. 7.
   Regarding rainfall, in our intention we do not use incorrect rainfall, we use the best available modelled rainfall. Please refer to the answer to your general comment 1.

8. If we look at figures 7 and 8, we can see that WRF-hydro does a really bad job (Figure 8) in basins where WRF precipitation has good performance (Figure 7), can you explain why?
   Fig. 7 and Fig. 8 are now Fig. 8 and Fig. 9. WRF-Hydro forced with modelled rainfall seems to poorly simulate especially watersheds Af (Nov 1994), Pg (both events), Le (both events), Mk (Jan 1989), Li (Nov 1994), An (Jan 1989), Pl (Jan 1989). For An and Pl, we believe that the problem is the same as for observed rainfall, therefore related partly to the difficult parameterization of a highly fractured bedrock and partly to the high elevation causing snow. Watershed Af, for Nov-1994 event show a very high rainfall PBIAS. Watershed Le shows rather low rainfall NSE for both events and a very high rainfall PBIAS for Jan-89 event. Watershed Li show a medium-high MAE for Nov-1994. Watersheds Pg and Mk are those characterized by the lowest average discharge during both events. Therefore, small discharge variations cause higher performance loss for them than for all other watersheds. They are the perfect exemplification of what we wrote in the manuscript about the small shifts in the space-time rainfall fields causing important performance losses. Figures S3 and S4 in the supplementary material show it. We have modified the manuscript as follows to stress this point.

   **Results, section 5.3 WRF-Hydro simulations with modeled precipitation, Line 470-475:**
   "These results indicate that a small shift in time or space of modelled rainfall, in comparison to

observed precipitation, can strongly modify the hydrologic response of small watersheds to extreme events. This is particularly evident in watersheds Pg and Mk, which are among the smallest and those characterized by the lowest average discharge in both events (Fig. 6, Fig. 7, Fig. S3, Fig. S4). Although their rainfall performance indices (Fig. 8) do not show particularly large errors (except a negative NSE for Mk in Nov 1994), streamflow fit indices present very negative values and streamflow PBIAS is very high as well (Fig. 9)".

9. In the conclusion you mention this "Streamflow obtained with WRF-modelled rainfall forcing showed high discrepancies with observations, despite the good agreement between modeled and observed precipitation (average NSE of 0.83 and 0.49 for Jan1989 and Nov 1994, respectively)." Did you try to calibrate the model with the modeled precipitation data and evaluate the observation?
This is a good point. We thought about it during the design of the methodological approach but then we preferred to focus on the performance loss passing from a calibration on observed rainfall to a simulation with modelled rainfall. Also, we are aware that the observed gridded dataset could be characterized by errors but the high density of stations and the study done on the creation of the daily dataset, including the evaluation of the final product (Camera et al., 2014), gave us some confidence about the quality of the input data.

**References not present in the manuscript**
Simmons, A, Soci, C, Nicolas, J, Bell, B, Berrisford, P, Dragani, R, Flemming, J, Haimberger, L, Healy, S, Hersbach, H, Horányi, A, Inness, A, Munoz-Sabater, J, Radu, R, Schepers, D, 2020. Global stratospheric temperature bias and other stratospheric aspects of ERA5 and ERA5.1. ECMWF Technical Memoranda 859, Reading (UK), 40 pp.

---

## Author Comment (AC3) · 19 Jun 2020

Please refer to the attached pdf for all updated figures.

Please also note the supplement to this comment:
https://www.nat-hazards-earth-syst-sci-discuss.net/nhess-2020-43/nhess-2020-43-AC3-supplement.pdf

———————————————————

2020-43, 2020.

---

## Author Response (AR2)

**Simulation of extreme rainfall and streamflow events in small Mediterranean watersheds with a one-way coupled atmospheric-hydrologic modelling system**

Corrado Camera[1], Adriana Bruggeman[2], George Zittis[3], Ioannis Sofokleous[2], and Joël Arnault[4]

[1] Dipartimento di Scienze della Terra 'A. Desio', Università degli Studi di Milano, Milano, 20133, Italy
[2] Energy Environment and Water Research Center, The Cyprus Institute, Nicosia, 2121, Cyprus
[3] Climate and Atmosphere Research Center, The Cyprus Institute, Nicosia, 2121, Cyprus
[4] Institute of Meteorology and Climate Research, Karlsruhe Institute of Technology, Garmisch-Partenkirchen, 82467, Germany

*Correspondence to*: Corrado Camera (corrado.camera@unimi.it)

**Answers to reviewers**

We would like to thank both reviewers and the editor for the time they took to go through the second version of our manuscript and our answers to their comments. Following, we provide answers to the new minor comments and the third manuscript version with track changes respect to version two.

**Anonymous Referee # 1**

No new comments

**Anonymous Referee #2**

1. I am still having problems with this conclusion "This suggests that model calibration with modeled rainfall forcing is not optimal for small mountain watersheds and should be carefully evaluated if no other options are available" I think that the performance would be bad if the inputs are bad (observed or modeled data). This is not a work on WRF so there is no information except for the evaluation of 1 downscaled product. What if the forcing used to run WRF was incorrect? Maybe if ERA5 was used or another reanalysis, you could have gotten better results. So I suggest avoiding these statements since you actually don't provide enough information to make this generalization.
   **Answer:** We agree with the reviewer that the downscaled product used in our study is just one and it is therefore impossible to generalize the result. We have deleted the sentence suggested by the reviewer and the following one, too.

[revised manuscript text omitted]